# Seed-Fill-Shift-Repair: A redistricting heuristic for civic deliberation

Christian Haas[1], Lee Hachadoorian[2], Steven O. Kimbrough[3]*, Peter Miller[4], Frederic Murphy[5]

**1** College of Information Science and Technology, University of Nebraska, Omaha, NE, United States of America, **2** Geography and Urban Studies, Temple University, Philadelphia, PA, United States of America, **3** Operations, Information and Decisions, University of Pennsylvania, Philadelphia, PA, United States of America, **4** The Brennan Center for Justice, New York University School of Law, New York, NY, United States of America, **5** Fox School of Business, Temple University, Philadelphia, PA, United States of America

* kimbrough@wharton.upenn.edu

**Data Availability Statement:** All data and source code are publicly available as indicated in the paper (via DOIs on Zenodo). Our source code is available in the Supporting Information and via Zenodo. Data

## Abstract

Political redistricting is the redrawing of electoral district boundaries. It is normally undertaken to reflect population changes. The process can be abused, in what is called gerrymandering, to favor one party or interest group over another, resulting thereby in broadly undemocratic outcomes that misrepresent the views of the voters. Gerrymandering is especially vexing in the United States. This paper introduces an algorithm, with an implementation, for creating districting plans (whether for political redistricting or for other districting applications). The algorithm, Seed-Fill-Shift-Repair (SFSR), is demonstrated for Congressional redistricting in American states. SFSR is able to create thousands of valid redistricting plans, which may then be used as points of departure for public deliberation regarding how best to redistrict a given polity. The main objectives of this paper are: (i) to present SFSR in a broadly accessible form, including code that implements it and test data, so that it may be used for both civic deliberations by the public and for research purposes. (ii) to make the case for what SFSR essays to do, which is to approach redistricting, and districting generally, from a constraint satisfaction perspective and from the perspective of producing a plurality of feasible solutions that may then serve in subsequent deliberations. To further these goals, we make the code publicly available. The paper presents, for illustration purposes, a corpus of 11,206 valid redistricting plans for the Commonwealth of Pennsylvania (produced by SFSR), using the 2017 American Community Survey, along with descriptive statistics. Also, the paper presents 1,000 plans for each of the states of Arizona, Michigan, North Carolina, Pennsylvania, and Wisconsin, using the 2018 American Community Survey, along with descriptive statistics on these plans and the computations involved in their creation.

## Introduction

A *districting problem* (also known as a *zone design problem*) is present whenever there are a number of *atomic areal units* (AAUs) that must be assigned to one of a smaller number of

and ancillary files are available from Zenodo, as indicated in the paper.

**Funding:** The author(s) received no specific funding for this work.

**Competing interests:** The authors have declared that no competing interests exist.

aggregations or *districts*, subject to constraints [1, 2]. (Hereafter, we use *units* in general discussion and *precincts* in the context of our algorithm, SFSR.) Districting problems are computationally challenging and have been shown to be computationally complex [3–5].

Democracies routinely face districting problems when they undertake *redistricting*, the revising of the boundaries of legislative districts in response to population changes [6, 7]. Redistricting is an especially important activity in the United States, where it is required after every decennial census in order to maintain population equality among the 435 electoral districts in the lower house of Congress, legislative districts in the States, and local governing bodies, most of which are elected among single-member districts and on the basis of a majoritarian vote. The American redistricting process has become highly contentious politically, as an authoritative monograph on the subject declares in its title *Redistricting: The Most Political Activity in America* [8], giving rise to the common term gerrymandering as a catch-all for manipulation of electoral procedures for some sort of gain. (While MacDonald [9, 479] observes "The only thing that the average person seems to know about redistricting is the term 'gerrymander,' a concept that is almost always misunderstood" the U.S. Supreme Court defined gerrymandering as "the drawing of legislative district lines to subordinate adherents of one political party and entrench a rival party in power" (*Arizona State Legislature v. Arizona Independent Redistricting Commission* 576 US __ (2015)).) The redistricting process in the United States is also comparatively exceptional. "Compared with other countries with plurality and majority systems, the American redistricting process is much more political . . . Most other countries have neutral commissions to draw the district boundaries. These exist in some American states as well, but the usual pattern is for the state legislatures to be in charge" [10]. This power to dictate the course of the redistricting process can have an effect on the outcome of elections, and therefore policy outputs [11], especially when one party controls a legislative majority and holds the governor's office, rendering the minority party unable to play any meaningful role in the districting process. As the famous or infamous, depending on your perspective, Republican redistricting consultant Thomas Hofeller once observed "Redistricting is like an election in reverse. It's a great event. Usually the voters get to pick the politicians. In redistricting, the politicians get to pick the voters." (https://www.npr.org/transcripts/730260511 (last accessed March 4, 2020).)

Redistricting is straightforward when one political party controls the process. The goal is to maximize the number of seats won by that party [12]. It is well understood how to do this; the U.S. Supreme Court defined "stacking" (since supplanted by the term "packing") and "cracking" as two ways to produce partisan advantage in districted elections (*Davis v. Bandemer* 478 US 109 [1986]). (See footnote 6 of the decision: "These are familiar techniques of political gerrymandering. Democratic (or Republican, as the case may be) votes are 'stacked' and 'wasted' by creating districts where Democrats form majorities much greater than the 50% necessary to carry those districts. Concurrently, Republican votes are spread among districts in which they form safe, perhaps 55%, majorities, and Democratic votes are 'cracked' or 'split' by dispersing them in such a way as to be ineffectual.") Producing a plan that meets the interests of voters is far more difficult because the goals of such a plan are less well defined and different segments of the population emphasize different criteria.

Nonetheless, we envision the algorithm we describe here as one means to disrupt the conflict of interest inherent in the redistricting process when legislators are drawing districts: an objective to increase the likelihood of their own reelection and the endurance of their party's majority. Laboratory experiments show that impartial advice can reduce selfish behavior in a dictator game [13]. We view the simulated district maps produced here as a form of impartial advice to legislators designed to reduce the partisan aspect of redistricting, in line with a similar study of Japanese parliamentary districts which produced computer-based districts closer

**Table 1. Fifty precincts, 30 Cyan party, 20 Yellow party.**

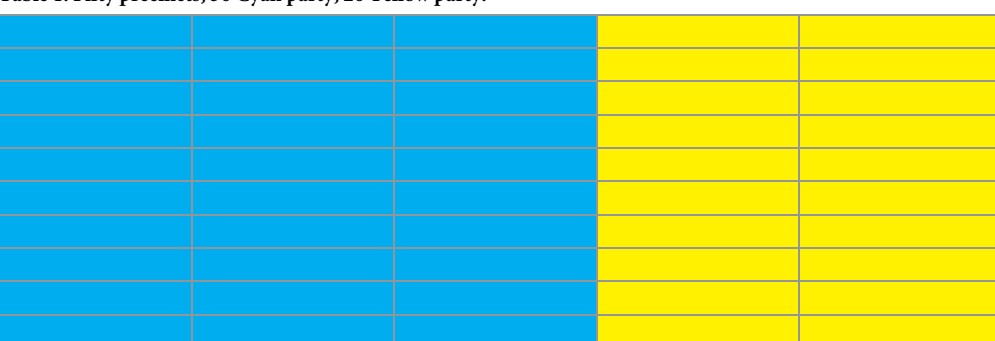

Fifty voting precincts are to be apportioned into ten contiguous districts. Each precinct is dominated by either the Cyan or the Yellow party and each precinct has approximately the same voting population.

to population balance when compared to the plan drawn and implemented in 1994 [14]. At the same time, these alternative maps can serve as a locus of public deliberation among private individuals, and interest group representatives, when selecting among a set of feasible districting plans.

   Gerrymandering may be characterized as abuse of power for achieving advantage in redistricting. This advantage is typically along partisan, racial or ethnic, or incumbency lines. Tables 1–4 graphically illustrate how this can be done. Where the lines are drawn to delineate districts can affect the partisan outcome of elections, even with population equality among districts and contiguity within them. Table 1 presents a baseline case. There are 50 atomic areal units (voting *precincts* hereafter), which must be assigned to 10 districts. There are two parties, the Cyan party and the Yellow party. Each precinct is dominated by one or the other party, and each precinct is equal to the others in population, so each of the 10 districts should have 5 precincts. Finally, the Cyan party, with 30 precincts, represents the majority of the population. Although this is a highly stylized representation, the principles that can be illustrated with it are quite realistic and are widely realized [15–17].

**Table 2. Fifty precincts, 30 Cyan party, 20 Yellow party.** Ten districts, 6 fully Cyan, 4 fully Yellow.

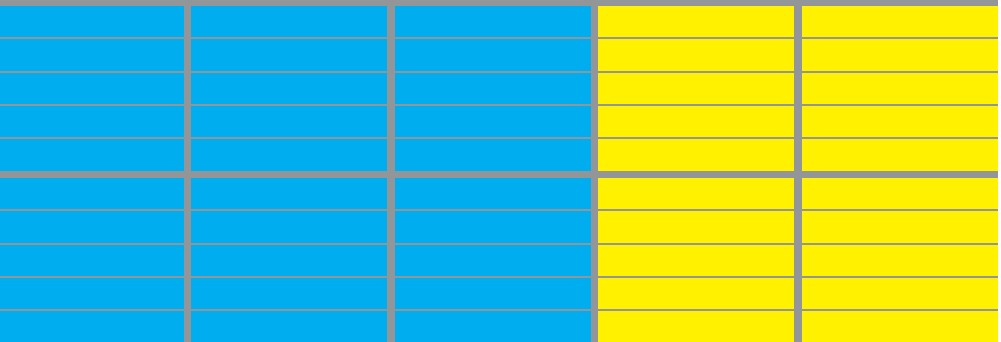

Fifty voting precincts are to be apportioned into ten contiguous districts. Each precinct is dominated by either the Cyan or the Yellow party and each precinct has approximately the same voting population. In this plan, the percentage of Cyan (Yellow) districts equals the percentage of Cyan (Yellow) precincts. No partisan bias is evident in the plan. All districts are homogeneous by party.

**Table 3. Fifty precincts, 30 Cyan, 20 Yellow.** The ten districts all favor of Cyan, which has 60% of the vote in each district.

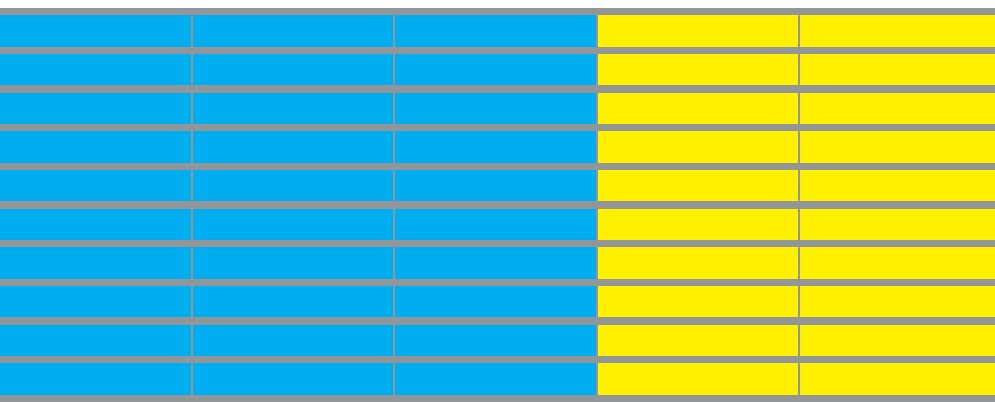

Fifty voting precincts are to be apportioned into ten contiguous districts. Each precinct is dominated by either the Cyan or the Yellow party and each precinct has approximately the same voting population. In this plan Cyan has a strong majority in each of the 10 districts, suggesting a partisan gerrymander in favor of Cyan.

Table 2 imposes a (re)districting plan upon our reference polity, Table 1. The Table 2 plan is not a gerrymander, at least for partisan advantage. The Cyan (Yellow) party has 60% (40%) of the voters and dominates in 60% (40%) of the districts.

In the plan of Table 3, all 10 districts have a comfortable 60% majority for the Cyan party, which could be be expected to take 100% of the seats, that is, all ten of them, leaving none for the Yellow party even though its voters constitute 40% of the population. This plan, in consequence, embodies a partisan advantage favoring the Cyan party. It would normally be considered a partisan gerrymander, something the Cyan party would do if it had the power.

**Table 4. Fifty precincts, 30 Cyan, 20 Yellow.** Of the ten districts, 4 are fully Cyan, 6 have a Yellow majority, suggesting a gerrymander in favor of Yellow.

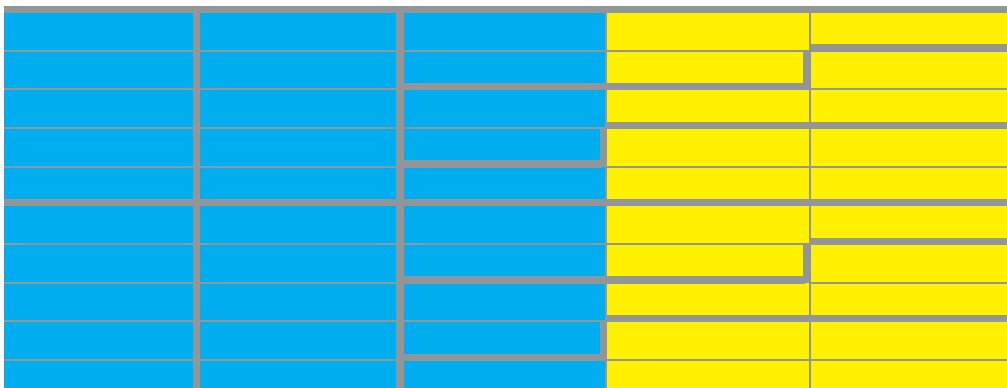

Fifty voting precincts are to be apportioned into ten contiguous districts. Each precinct is dominated by either the Cyan or the Yellow party and each precinct has approximately the same voting population. This plan evidences the two basic gerrymandering techniques: "packing" and "cracking." The Cyan party has suffered both. It is packed into 4 districts, creating seats effectively uncontested by the Yellow party. The remaining 10 Cyan precincts are cracked (split) across 6 districts in order to put them into a minority in each case. In consequence, while Yellow has 40% of the voters, it has majorities in 60% (6 out of 10) of the districts.

Table 4 shows how a party can create a partisan majority among the districts even if the party does not have a majority of the voters, i.e., even if a majority of the voters favor the other party.

Publicly-oriented redistricting criteria, numbering in the dozens, have appeared in the literature without a consensus on which criteria are most important or even substantially relevant. Traditionally, however, compactness, contiguity, and population balance have long been seen as important [18]. Even so, they are not without their issues.

Informally, a compact district is one in which the people in it live comparatively near each other. A district that is not compact is spread out, resulting in generally greater distances between people (keeping the number of people constant). A circle, it is widely agreed, is the most compact geometric shape for a district. Dozens of mathematically precise distinct definitions of compactness have been proposed, but none has received general assent (see Young [19] for a review). In the early Republic, compactness would have been a proxy for minimizing travel times between representatives and constituents, and compact districts were a requirement of the Apportionment Acts of 1901 and 1911. Compactness requirements and other "diversifying" criteria tend to reduce participation in elections [20]. However, there are now so many mathematical definitions of compactness that arguably the search for *the* measure of compactness is misguided [21]. Surveys of compactness measures include [19, 21–23]. Should the focus be on smooth district perimeters or should it be on coming close to circles or squares or some tradeoff of both? Young [19] has shown how the different definitions of compactness can be gamed. Because of the lack of agreement, Kaufman et al. [24] and Chou et al. [25, 26] have used surveys to observe how people value the different aspects of compactness.

Contiguity is perhaps the districting criterion that is easiest to define, but even here there are issues. Some states say a district is contiguous if two parts touch only at one corner point ("queen contiguity") while others disallow this and require that subareas in a district have overlapping boundaries ("rook contiguity") [27, 28]. Consider two precincts separated by water. People agree they are contiguous if a small river lies between them. A large bay that must be crossed by a circuitous route to go between two precincts probably should mean the two precincts are not adjacent. The same should apply to mountain ranges. Computing power has contributed to the frequency of districts drawn with "questionable contiguity" [29, page 343].

Population equality has its own problems. First, the decennial census is not a perfect count. Undercounts tend to be of renters, young males, immigrants, and African-Americans, and overcounts of homeowners, middle-aged females, and college students [30, 31]. Births, deaths, and migration between the census date and the release of redistricting data mean that even if an exact count were possible, it would no longer be exact when used for redistricting. Thus, requiring exactly equal population means being precise about an imprecise number.

There are follow-on consequences of these criteria. The tighter the population balance requirement is, the less compact the districts are, all things being equal [32]. Further, minimizing the number of split political units, e.g., the number of counties that are in more than one Congressional district or the number of school districts that are in more than one state senate district, is widely seen as an important criterion (often required by state law), yet the more restrictive the allowable population deviation, the more split political units there tend to be.

It is frequently the case that these criteria conflict [33]. People in rural areas and small towns, for example, perhaps strongly identify with local territories (spatially characterized social, governmental, and administrative units) more than residents of large cities. Rural areas are typically more dependent on state government for services. Thus, they are deeply concerned about losing voice when their local jurisdictions, particularly counties, are split when

defining state legislative boundaries [8, page 97]. People in large cities tend to be more mobile and may be more concerned about other criteria.

In addition to the four criteria we have discussed (contiguity, population equality, compactness, and splits), other criteria have been proposed. These include:

- Partisan fairness—Measures vary, but include the difference between median and mean partisans [34], the rate of wasted votes across the major parties [35–38], or the symmetry of seat shares if parties receive the same share of the vote [39–41].

- Preservation of existing political communities—Turnout is lower in areas that are "carved out" and placed into a different district [42, 43]. Furthermore, residents testifying at redistricting public comment hearings tend to advocate for keeping geographic units in a single district when possible [44, 45].

- Racial fairness—This is generally understood to be the creation of minority-majority districts when the minority population is large enough and sufficiently compact to draw a district, and racially-polarized voting is present, as articulated by the Supreme Court in *Thornburg v. Gingles* 478 US 30 (1986). See also [46].

Many other redistricting criteria have been actively discussed in the literature. See [8, 47, 48] for comprehensive but hardly exhaustive lists.

The implications of the state of this literature are twofold. The first is that the legitimate criteria for redistricting constitute a dauntingly large and challenging collection with no clear winners. Some of the criteria are naturally treatable as constraints because they are universally legally binding, particularly contiguity and population balance. The others are naturally conceived of as goals that belong in an objective function, or multiple objective functions, creating in either case computationally problematic optimization models. In general these models are solved using heuristics that do not guarantee optimality. The second consequence is that there does not exist anything like general agreement as to which criteria, under what definitions, and with what principles of mathematical combination (e.g., weights in the simplest case of linear objective functions) should constitute an objective function for redistricting. So, even if the optimization models were solvable, we do not know which of that indefinitely large set we should solve and present for public deliberation.

Fully recognizing this difficult situation, we propose a different kind of approach for computational discovery of redistricting plans. Our proposal is that for the sake of subsequent deliberations we eschew an objective function entirely and seek multiple solutions to a constraint satisfaction representation of the problem. We propose to use contiguity and population balance (within a specified tolerance) as the operating constraints and then to find thousands or more solutions that satisfy these constraints. These feasible solutions can be expected to be legally valid in most jurisdictions, although they cannot be expected to be especially good on any of the redistricting criteria. Our philosophy is one of *solution pluralism* [26, 49].

By solution pluralism, we mean the development of an appropriate model, in this instance a constraint satisfaction representation for redistricting, which we then use with an algorithmic, computational solver to generate a plurality of solutions to the model. The main technical contribution of this paper is to introduce the Seed-Fill-Shift-Repair algorithm for this purpose, which is able practicably to produce tens of thousands of redistricting plans that are feasible with respect to population balance and contiguity. The consideration set of plans can be presented for public deliberation and refinement. Our view is that the corpus of plans should constitute a point of departure for deliberation, with plans being modified in the direction of valued criteria and final choices being made based upon the resulting expanded set. For

researchers who wish to construct their own plans, including biasing the search for plans using various objectives, we make the code publicly available.

The remainder of this paper focuses on articulating and describing the Seed-Fill-Shift-Repair (SFSR) algorithm. The "Related work" section describes previous approaches, mainly optimization approaches, to computational discovery of redistricting plans. The "Methods" section presents the Seed-Fill-Shift-Repair algorithm in detail. The "Computational results" section discusses selected computational experience with the algorithm, for purposes of illustration, although the data are real. The "Discussion" section presents various reflections arising from the availability of SFSR. The "Conclusion" section indicates future research and paths forward for practical use of the algorithm.

## Related work

The portion of the literature on redistricting we cover here has two main themes. The first is to "optimize" the districting plans, initially trying to find the most compact map while satisfying population and contiguity constraints. The second is using simulation techniques to generate plans. (Our discussion below references peer-reviewed, academic studies of the subject. We mention here, however, three notable open source and web-based processes that have not yet been described in published studies. Auto-Redistrict http://autoredistrict.org/ is a multi-attribute genetic algorithm that draws fair and compact districts. BDistricting https://bdistricting.com/2010/ draws compact and population balanced districts. Antimander https://antimander.org/ uses a genetic algorithm to create districts and allows the user to see the trade-offs between various criteria in each plan.) The simulation literature has taken two directions, defining measures that could be used by courts to decide if districts have been gerrymandered and developing methods for randomly constructing large numbers of districts that can be used to compare a state's districts against a distribution of redistricting plans on different measures.

We should note at the outset that neither approach has had a great deal of uptake in actual redistricting procedures, although some of the work has appeared in court proceedings. Political leaders have tended to resist using independent commissions and publicly available analytical techniques. Birge [50], for example, used quadratic programming to minimize the splitting of political subdivisions. The Michigan Supreme Court then chose a plan with more split subdivisions that favored the party of the majority of judges. (See *In re Apportionment of State Legislature* 413 Mich. 96, 321 N.W.2d 565 (1982).)

Optimization was the first technique used to redress gerrymandering formally. Hess et al. [51] and Weaver and Hess [52] describe the first successes in the use of optimization for redistricting. The original Hess et al. [51] study used heuristics to find the most compact plan among the plans generated. They randomly picked centers for districts and minimized the sum of centers of gravity around the districts, thereby favoring the objective of compactness. Using the generated map, they calculated new centers of gravity and then re-optimized, repeating until the centers stabilized. For small planning problems it is possible to enumerate all possible solutions [53]. Somewhat larger problems are amenable to exact algorithms. Garfinkel and Nemhauser [54] used implicit enumeration to find the most compact solution. Openshaw [55] addressed the size problem by first aggregating political units and then completely enumerating the possible plans. Taylor and Gudgin [56] dealt with the issue of the spatial distribution of voter affiliations, constructing frequency distributions of voter shares and generating by a simulation method all plans for 99 counties combined into 6 districts.

The problem with trying to find an exact solution is that, as the number of districts and population units (the term precincts in the main terminology of this paper) increases, the solution time suffers from combinatorial explosion. If, for the sake of argument, we take advantage

of the multitude of measures of compactness as a proxy measure for a "good" district, finding an exact solution is less important than a good solution. Thus, heuristics have been used to solve larger problems, e.g. Hess et al. [51] and Nagel [57]. Macmillan and Pierce [58] used the simulated annealing heuristic to find compact solutions for the small states of New Hampshire and Maine. Bozkaya et al. [59, 60] applied a search heuristic termed tabu search, which has performed well on many combinatorial problems. Gopalan et al. [61] used a restricted set of locations to serve as the centers of districts while using census tracts and their populations to accumulate voters in districts.

Although compactness is the most common objective in optimization models of redistricting [62], as noted above Birge [50] minimized splits. It is possible to use any mapping criterion or a weighted combination of criteria as the objective function. An interesting case is [63], which uses simulated annealing on both a single objective redistricting problem and a multi-objective redistricting problem in which both population equality and compactness served as objectives.

One of the problems with taking a standard optimization approach in redistricting has been that the number of contiguity constraints explodes with the size of the problem and impedes practical solution. Important progress has been made recently in this regard [64], but serious difficulties remain. Consequently, the models tend to be solved without the contiguity constraints then solutions are repaired after optimization. More fundamentally, in optimization the algorithms and heuristics find one optimal or good solution as if the objective were a precise measure of goodness. That there are so many definitions just of compactness means that the true objective for a plan is not as precise as a mathematical formulation implies. Consequently, one would like to have multiple solutions that are almost as compact as the most compact so that more generalized measures of goodness can be used to select or evaluate a plan. This also applies to splits and any other criteria of import, such those discussed above.

Recognizing the value of multiple solutions has led many researchers to use simulation techniques to generate large numbers of plans that are perceived as representative of the set of all possible plans and use those to assess plans coming from legislatures. The early literature mainly consists of *ad hoc* methods for generating low numbers of plans. Over time the sampling methods have become more sophisticated and the sample sizes have grown dramatically. We have ordered the literature review by year to highlight that the early papers had small samples and no real measures of statistical reliability. Over time the sample sizes have grown by many orders of magnitude and there has been some attention to how statistically meaningful the results are.

Vickrey [65] proposed using randomly generated plans to test for gerrymandering. Nagel [57] is an early implementation of automated redistricting. She starts with a solution, then improves it by swapping precincts on boundaries, and repeats the steps three times. Thoreson and Liittschwager [66] produced 150 plans by using simple greedy heuristics after randomly selecting initial starting points. They repeated this process for 3 different starts. Engstrom and Wildgen [67] generate 165 plans but their paper does not provide a description of their method. Openshaw [55] aggregates blocks of voters to reduce the number of possibilities and then enumerates all possible maps. Taylor and Gudgin [56] generated plans for the 99 counties in Iowa combined into 6 districts. O'Loughlin and Taylor [68] uses the Weaver-Hess and Nagel algorithms to generate 100 feasible solutions. O'Loughlin and Taylor also used Weaver and Hess's algorithm and Nagel's algorithm to generate 180 feasible solutions when examining redistricting in Mobile Alabama.

By the 2000s, increased computing power made it possible to generate many more plans. Cirincione, Darling, and O'Rourke [69] applied four different algorithms to generate plans that are biased towards four different criteria, using random starting points. The first

algorithm consists of selecting one block, adding from the perimeter randomly, repeating until population equality is achieved. This guarantees contiguity. The second algorithm focused on compactness. They place a box around a district and select from within the box, repeating until population equality is achieved. Third, to minimize splits of political districts they select from blocks in the same counties as the starting points until none are left. The fourth combines compactness and county rules. Altman and McDonald [70] have produced an open source district generator. As others have done since, it is suitable for helping individuals to construct plans interactively online through a web browser. (See https://www.tides.org/tides-news/media/tides-awards-2013-pizzigati-prize-to-fair-elections-pioneer-micah-altman/.)

Chen and Rodden use a heuristic optimization procedure, after freezing the existing minority majority districts, that is aimed at finding compact districts satisfying population balance and contiguity constraints [71, 72]. In their simulation of Florida and in later simulations, their procedure starts with a set of building blocks consisting of political jurisdictions that form the atomic units, in this case counties and cities, thus aggregating the smaller units used in actual redistricting. They sample one element from the set, measure the distance from the centroid of this element to the centroids of adjacent elements, and combine the sampled element with the closest adjacent element. This becomes a new element in the set, replacing the two combined elements. The process is repeated until the number of elements in the set is reduced to the number of voting districts. There is no attempt to achieve population equality in this phase. They then apply a repair algorithm that moves building blocks on boundaries from overpopulated districts to underpopulated districts. They produce 1000 districting plans and calculate their convexity scores, measuring convexity by taking the ratio of area of each district and the area of the convex hull of that district and summing over all districts. They then compare the compactness of the Florida plan in their paper [71] against the distribution of compactness in their sampled plans and find it to be a statistical outlier [73].

Chen and Cottrell [74] estimate the partisan gains across the country through gerrymandering by generating multiple districts for all states and comparing the simulation results with the actual outcomes. Their simulation algorithm is as follows. In each state they randomly place a seed in one of more than 15k squares, then add adjacent squares until the district reaches the population required. They repeat this 27 times for all Congresspersons. If at any point the map cannot be completed, it is discarded and the algorithm restarted. There is no repair process. They generate 200 maps for each state. Their estimate is 5 Republican seat gains in Republican controlled states, 3 Democratic gains in Democratic controlled states and 1.75 Democratic gains in the states subject to pre-clearance under the voting rights act.

Mattingly and Vaughn [75] sample redistricting plans for North Carolina using a Markov Chain Monte Carlo simulation to move from one redistricting plan to another. To achieve statistical independence among the plans they use in their analysis, they save a plan only after 100,000 steps in the Markov chain. Each plan has a weight that is a convex combination of its compactness and population equality scores, allowing the authors to examine the trade-off between compactness and population equality. Fifield and colleagues [62] describe a similiar MCMC method applied to data from Pennsylvania. This paper also begins a line of research comparing the variety of redistricting algorithms by observing that the "standard" seed-type models fail even under no population constraint. Tam Cho and Liu [76] use more finely grained political units/precincts than Chen and Rodden, generating over 1 billion plans on a parallel computer, of which they keep 250,000. They do not detail their algorithm, but do in a subsequent paper. (The question of whether these simulation algorithms must be public arose in the decision in the partisan gerrymandering case *Ohio A. Philip Randolph Institute v. Householder* 373 F. Supp. 3d 978 (S.D. Ohio 2019), see footnote 360.) Liu, Tam Cho, and Wang [77] generate 1 billion plans by placing random seeds along the border of a state then add adjacent

geographical units to randomly grow 200 maps. (They also consider random seeds throughout a state.) Then they mutate maps by switching adjacent units, either one or several at a time. Magleby and Mosesson [78] convert a map into a graph with the nodes having the population of the geographic unit and arcs representing adjacent units. After aggregating the graph, they randomly select a node as a seed then randomly select adjacent units until the population grows to the required level. Then a new seed is selected and the process repeated until it is clear that districts are formed or it is no longer possible to satisfy the contiguity requirement.

A new literature is developing that tries to understand the properties of the generated plans that use randomized algorithms. Wang [79, 80] proposes three statistical tests to use with simulated outcomes.

Altman and McDonald [81] generated 1 million plans and threw out 850,000. They found about 17 new plans for every 20 rejections at the one millionth iteration, showing that there are a vast number of plans that could potentially be generated. They find that plans do not look as if they are randomly generated. With their map, more vertically oriented districts are generated than horizontally oriented districts. (This is a curious outcome that needs investigation. Perhaps because their graph has more vertical adjacencies, it can bias the sampling by frequency of vertical and horizontal adjacencies. The problem is that when selecting a block from a cluster, a small block is more likely to be taken than a large block because the cluster of small blocks has more adjacencies. That is, all districts seem to absorb cities and towns before rural areas. A random radial direction rather than adjacency would solve this problem. Another would be to weight the odds of taking a boundary district by the length of the common boundary.) These results raise the question of what randomness means in the context of generating districts.

Many of the MCMC methods use a transition process consisting of flipping units between adjacent districts to create new districts. Since a single Markov transition does not cause a major change in in the starting map, it takes many iterations before the process produces a distinctive map that is statistically independent of the starting map. Consequently, many steps in the random walk must be discarded before retaining a plan. DeFord, et al. [82] propose a new random walk called recombination (ReCom). Paired adjacent districts are combined, with a bias toward selecting pairs with longer shared borders. A random spanning tree (set of edges that visit all nodes without creating a cycle, or possibility of revisiting the same node without reversing direction) is created, and then an edge is deleted that splits the spanning tree into two equal population districts (within the specified tolerance). They qualitatively show that the flip-based random walk still demonstrates memory after 1,000,000 steps (that is, the plan can be visually seen to be a transformation of the seed plan), while ReCom shows no memory (that is, generates independent plans) after 100 steps or fewer.

The problem of determining whether a simulation generates a representative sample is an area of ongoing research and disagreement. It is well understood that Markov chains converge on a unique stationary distribution, but it may not be possible in any specific application to characterize the stationary distribution, nor to compute the number of steps necessary ("mixing time") to approach this distribution within a specified threshold. Researchers must therefore rely upon heuristic convergence tests [82]. Chikina, et al. [83] prove that a reversible Markov chain can generate a global $p$-value at a discount (roughly the square root) of the outlier cutoff of any random walk from the Markov chain. Tam Cho and Rubinstein-Salzedo [84] argue that overly tight constraints in the chain can leave the algorithm exploring a space disconnected from the global state space. In their view, finding a local outlier can never allow claims regarding global outliers.

We are concerned whether the heuristics alluded to [82] can adequately characterize the global distribution. We contend that this problem is basically insoluble: the sample space is

effectively unknowable as the number of possible plans is roughly the number of combinations of $k$ districts taken from $n$ blocks (ignoring contiguity and population equality) and precisely it is a Stirling number of the second kind. Tam Cho and Liu [76] estimate that the number of possible allocations of 55 units to 6 districts is $8.7 \times 10^{39}$. The number of possible districting plans for larger states is explosively larger. This count has to be tempered because almost all randomly generated plans violate contiguity. Given that much of the recent literature focuses on using randomly generated plans to evaluate the likelihood of an official districting plan being unbiased or biased, the sample has to be representative of the population of plans and we do not fully understand what representative means.

We conclude this look at the related work by briefly noting a number of other automated approaches arising in the operations research and metaheuristics literature.

Forman and Yue [85] innovatively formulate Congressional redistricting problems as augmented traveling salesman problems and use a genetic algorithm metaheuristic to find plans that in the objective minimize a function of deviation from population balance, deviation from contiguity, and deviation from ideal compactness. This is augmented by post-processing to improve on population balance. They report success in finding contiguous plans with a population balance of within 1% of ideality and that are strongly compact on the Schwartzberg measure. As reported, their approach only works well for small states with six or fewer districts.

MacMillan [86] develops a districting algorithm, called SARA, based on simulated annealing. The focus of the paper is on fast contiguity checking when a precinct ("zone" in the terminology of the paper) is considered for removal from a district. The paper introduces such a method and compares it with a prior method. The paper does not report results on producing districting plans. Interestingly, the proposed simulated annealing formulation is an unconstrained optimization seeking to minimize population deviations among districts. SARA commences with a contiguous plan and enforces contiguity during its operation. In consequence, it can be interpreted as effectively a constraint satisfaction algorithm, as is SFSR, but with a very different procedure and without reporting districting results. See Browdy [87] who also uses simulated annealing on redistricting problems.

Caro et al. [88] use binary integer programming to do redistricting for Philadelphia schools —rather than political redistricting, which is our focus—with the objective of minimizing student travel distances within neighborhood clusters of schools. The formulation in the paper is simpler than the other redistricting formulations presented here and is unlikely to work for Congressional redistricting for at least two reasons. The first is scale. The approach was applied to individual clusters of a dozen or fewer schools that are existing buildings rather than a model where the centers are determined as part of defining districts. Second, the attributes of interest in Caro et al. are rather different from those of redistricting problems. For example, population balance is not considered and contiguity is not preserved. What is strikingly relevant to political redistricting, however, is the paper's characterization of the districting problem as being inherently "ill-defined," a point of immediate relevance to SFSR and one we address in the sequel.

> The application of mathematical programming to school redistricting problems certainly helps to understand the problem and generate rational solutions, but does not make the final decision. One reason for this is that a school redistricting problem is almost always ill-defined. While some districting criteria are relatively well suited for numerical treatment, others are too elusive to be quantified and may be overlooked (if not intentionally ignored). Thus, no matter how elegantly a school redistricting problem is formulated or solved as an optimization model, the generated solution usually cannot avoid objection or modification.

If this modification cannot be done smoothly, the overall value of the school redistricting system may degrade significantly. [88]

Ricca and Simeone [4] apply and compare five metaheuristics (Descent, aka hill-climbing, Tabu Search (in two forms), Simulated Annealing, and Old Bachelor Acceptance (a variety of threshold acceptance algorithms [49])) to several Italian regional redistricting problems. The focus of the paper is optimizing against three criteria, each to be minimized: population inequality, noncompactness, and nonconformity to administrative boundaries. The work is focused on comparing performance of the tested metaheuristics on aspects of the problem. The paper is especially useful for its treatment of the computational difficulties of the problem. Many of the ideas in the paper are relevant to post-processing (and improvement of) corpora of solutions found by SFSR.

See [5, 89, 90] for overviews of important earlier redistricting algorithms in this literature.

## Methods

Seed-Fill-Shift-Repair (SFSR) is best framed as a heuristic constraint satisfaction algorithm [91, 92], in contrast to an optimization algorithm. In optimization, we seek to maximize (or minimize) an objective function of the decision variables (for example a measure of compactness), subject to satisfying constraints on the decision variables. Abstract mathematical formulation is helpful for stating the distinction clearly. We may think of the general formulation of a constrained optimization problem as follows:

$$\max_{\mathbf{x}} z \quad = \quad f(\mathbf{x}) \tag{1}$$

$$\text{subject to} \tag{2}$$

$$g(\mathbf{x}) \quad \in \quad C \tag{3}$$

In the case of redistricting, the decision variables correspond to assignment of basic units (voting precincts, census tracts, etc., as the case may be; we default to *precincts* in this paper) to districts. Here, in a mathematical formulation, $\mathbf{x}$ would typically be a vector of individual decision variables, $x_{i,j}$, with the interpretation $x_{i,j} = 1$, if unit $i$ is assigned to district $j$ and 0 otherwise. We seek to set the values for all of the $x_{i,j}$s so that collectively they will maximize (or minimize) the objective function, $f(\mathbf{x})$, while at the same time satisfying the constraints, abstractly represented as $g(\mathbf{x}) \in C$. The values they may take on are constrained in the constraints, $g(\mathbf{x}) \in C$, to be in the set of districts for the instance of the model. We say that a *solution* to the problem is any complete assignment of values to the decision variables. A *feasible solution* is a solution that also satisfies the constraints. An *optimal* solution is a feasible solution such that no other feasible solution has a superior objective function value. (It is entirely possible that a problem has more than one optimal solution.)

The redistricting algorithms we are aware of, reviewed in the "Related work" section, generally aim in one way or another at optimization, e.g., to favor searches that find more compact solutions. As noted above, there does not exist any consensus regarding what the objective function should be for redistricting. SFSR was created to circumvent this immediate problem by being a constraint satisfaction algorithm and so eschewing the objective function entirely. In constraint satisfaction problems, there is no objective function and we seek only to satisfy specified constraints on the decision variables. With reference to the abstract constrained optimization representation above, we may represent an abstract constraint satisfaction problem

as follows:

$$\text{Find } \mathbf{x} \quad \text{subject to} \tag{4}$$

$$g(\mathbf{x}) \quad \in \quad C \tag{5}$$

This mathematical formalization of constraint satisfaction problems is conceptually useful. Because our algorithm is a procedural heuristic, however, the underlying representation is procedural, rather than mathematical. The decision variables in our implementation are represented as a column in a table held in memory (in a Python geopandas GeoDataFrame) where the rows correspond to the basic units and the columns include information on the IDs of the basic units, the district to which the basic units are assigned (-1 if not assigned), and the populations of the basic units, among other information such as geographical shape of the basic unit. This data structure is created in the initialization step and the program variable `assignmentsDF` (assignments GeoDataFrame), with every unit assigned at initialization to the -1 district. Thus, `assignmentsDF` and its descendants correspond to the $x_{i,j}$s in the mathematical formulation schema, above, as well as carrying the unit population information.

We shall be concerned with three constraints, each of which is widely agreed in the redistricting literature to be necessary for valid plans.

1. Complete and exclusive assignment.
   Every unit must be assigned to exactly one district. This constraint is met by construction in SFSR. Upon completion of the `seed()` and `fill()` procedures, which are called once at the beginning of every run after initialization, every unit is assigned to a valid district (and not to -1). Thereafter, all reassignments of units to districts involve a unit changing its district to that of an adjacent district.
   A complete and exclusive assignment of units to valid districts constitutes a *solution* or a *plan*, which may or may not be feasible, depending on whether the remaining two constraints are satisfied.

2. Population balance.
   Election districts, especially Congressional districts in the United States, are required by the courts to have equal populations within a given state, as determined by the most recent decennial census. (See the U.S. Supreme Court decision in *Karcher v. Daggett* 462 US 725 (1983). Population equality standards are relaxed for legislative districts, but deviation of more than 5% above or below an ideal is likely suspect, as the Court found in *Harris v. Arizona Independent Redistricting Commission* 578 US __ (2016).) In implementing this constraint, SFSR uses a parameter, `tolerance`, to determine whether a district in a plan is balanced. By default we set `tolerance` to 0.01 (1%), a conventionally accepted level [83]. We say that a district, *j*, in a plan is feasible with respect to population balance if

$$(1-t)\left\{\frac{1}{|J|}\sum_{j\in J}p_j\right\} \le p_j \le (1+t)\left\{\frac{1}{|J|}\sum_{j\in J}p_j\right\} \tag{6}$$

where *t* is the tolerance, $p_j$ is the population of district *j* in the plan, and where $|J|$ is the number of districts to be populated. Both *t* and $|J|$ are parameters for SFSR, embodied in the program variables `tolerance` and `numDistricts`.

We say that a plan is population balance feasible if every district in it is population balance feasible.

3. Contiguity.
A district is contiguous if from every basic unit in the district it is possible to reach every other basic unit in the district by transiting through neighboring basic units that are in the district. More informally, a district is contiguous if it is possible to travel between any two units in the district without leaving the district.
We say that a plan is contiguous if every district in it is contiguous.
The contiguity constraint is complex to represent mathematically, although it can be done [61, 93]. SFSR uses a computational procedure, described in the sequel, to determine whether the districts in a presented plan are all contiguous.

With the foregoing as background we now give a high-level description of SFSR. See section "The SFSR algorithm in detail" for a more granular account. Our code, referred to here and below, is available in the "Supporting information" S1 materials.

1. Prepare data.
Two input files are required for the procedure: a source shapefile with the geographic information for the precincts, and a data file with precinct IDs and their corresponding population. These files can either be provided directly, or in case of Census tract or blockgroup-based districting by using a separate built-in code module that allows for the direct download of the corresponding files from the Census API by the U.S. Census Bureau.

2. Initialize.
During initialization, the shapefile and data file are merged via common precinctIDs to form a data frame with geographical and population information. This information is used to calculate the neighbors for each precinct. By default, and in the results reported here, we use rook neighborhoods in determining neighbors. Queen or any other neighborhood definition could of course be used as alternative.
Afterwards, each precinct is assigned to the fictitious district -1. All actual districts are numbered in the sequence from 1 to the number of districts, an input parameter with program variable `numDistricts`.

3. Seed.
The first part of the `seed_fill()` procedure randomly picks `numDistricts` different precincts. The procedure assigns each precinct a unique value in the sequence 1, 2, . . ., `numDistricts`.

4. Fill.
The second part of the `seed_fill()` procedure randomly picks district IDs and adds adjoining unassigned (-1) precincts to the current district ID until all precincts are assigned. At the completion of `seed_fill()` a plan exists because every precinct has been assigned to a valid district. The plan is contiguous by construction. There is, however, very little chance that the plan is population balanced.
Vickrey [94] is widely credited with being the first to publish the seed-fill method of creating a contiguous redistricting plan in this fashion.

5. Shift while not in population balance.
The shifting procedures reassign precincts from overpopulated (underpopulated) districts to neighboring underpopulated (overpopulated districts). The core shifting procedure is called `shift()` in our implementation.

`shift()` randomly picks a single precinct on the border of a random district, identifies a neighbor with a different assigned district and reassigns the district of the precinct or the neighbor as appropriate.

The shifting procedures continue in this manner, using `shift()`s until every district of the plan is population balanced.

6. Check plan contiguity.

   The current plan is population balanced. The plan contiguity is checked. If the plan is contiguous, then a feasible plan has been discovered and the algorithm cleans up, writes a file to record the plan, and exits; otherwise SFSR continues, sending information to the repair procedure regarding which precincts are separated from the main body in each district.

7. Repair.

   The current plan is population balanced. Using information from the contiguity check, the `repairFor()` procedure iterates through the districts, repairing one at a time and checking it for contiguity. This continues until every district is contiguous.

8. The current plan is contiguous. The population balance is checked. If every district is within the population tolerance, then a feasible plan has been discovered and the algorithm cleans up, writes a file to record the plan, and exits; otherwise the algorithm continues at step 5. If any district is not within the population tolerance, then `shiftWhile` (step 5) is invoked on the plan and another round of shifting, checking for contiguity, and repairing ensues. This continues until a feasible plan is found.

## The SFSR algorithm in detail

Having described SFSR at a high level, we now discuss in detail three components of the algorithm. Full details are available in the commented code implementing the algorithm, provided in the "Supporting information" materials.

**Shifting.** Stated more carefully, `shift()` is called within `shiftwhile()` which repeatedly calls `shift()` until the ensemble is population balanced. One call to `shift()` changes the district assignment of at most 1 precinct, and possibly may not change the assignment of any precinct. `shift()` first picks a random precinct, $u_i$, with assigned district $d_i$ from the current plan and on its border with another district, $d_j$. It then randomly orders all of the neighbors of $u_i$ and processes them sequentially. If a neighbor, $v_j$ is assigned district, $d_j \neq d_i$, `shift()` considers two possibilities.

i. If the population of $d_i$ is greater than the ideal population and the population of $d_j$ is less than the ideal population, then `shift()` changes the assigned district of $u_i$ from $d_i$ to $d_j$ and returns the revised current plan.

ii. If the population of $d_i$ is less than the ideal and the population of $d_j$ is greater than the ideal population, then `shift()` changes the assigned district of $v_j$ from $d_j$ to $d_i$ and returns the revised current plan.

   Points arising:

1. The shift step in the code is realized by multiple calls to `shift()`, until the current plan is population feasible.

2. The ideal population of a district (`idealPop` in the implementation) is the total population of all the precincts, divided by the number of districts to be planned.

3. Define `popSlack` as `tolerance * idealPop`. This is the maximum allowable population deviation from the ideal population in a feasible district. Mathematically, `popSlack =`

$$t\left\{\frac{1}{|J|}\sum_{j\in J}p_j\right\} \tag{7}$$

4. If the largest population of a precinct is less than `popSlack`, we say that the *population slack* condition is satisfied.
   To illustrate, if the ideal population of a Congressional district is 750,000 and the tolerance is 1%, then so long as the population of the largest precinct is less than 7,500, the population slack condition is satisfied. We note that census tracts range in population around a mean of about 4,000 people. We find that SFSR works unproblematically with a 1% tolerance when designing Congressional districts using census tracts as basic units.

5. If the population slack condition is satisfied, then across multiple calls to `shift()` by the code ensemble implementing the shift step, any district that is feasible—that is population balanced within the tolerance—will remain feasible under repeated calls to `shift()`. This is because if the population slack condition is satisfied, then removing a precinct from a feasible district with population greater than the ideal will leave the district population feasible. Similarly, adding a precinct to a feasible district with population less than the ideal will also leave the district feasible.

6. A population-feasible district may nonetheless change in composition over multiple calls to `shift()`, while remaining feasible as the shift step executes. Such districts can and do serve as conduits of population moving from under or over populated districts on one side to over or under populated districts on the other. In consequence, multiple calls to `shift()` eventually leads to a population balanced (feasible) plan, at which point the shift procedures halt and control passes to the next step of SFSR.

7. Cycling is possible between feasible adjacent districts, but has not proved to be a problem in practice, as the drawing of a random $u_i$ in `shift()` affords likely escape.

**Contiguity checking.** Contiguity checking for a single district is performed by `checkDistrictContiguity()`. Plans in turn are checked for contiguity by checking each constituent district for contiguity. To check a single district for contiguity, the current assignment of precincts to that district is extracted along with the geometries of the precincts. Afterwards, the number of connected components within this district is calculated based on the geometrical (polygon) information for the precincts. If a district is contiguous, all precincts are connected and the number of connected components is 1. If some precincts are not connected, the number of connected components will be larger than 1. Hence, this number of connected components determines if the district is contiguous (1 connected component) or not (>1 connected components).

In the event that **D** is not contiguous `checkDistrictContiguity()` returns a list of precincts that do not belong to the largest connected component. Specifically, a list of precinct and component IDs is part of the contiguity calculation, which then is translated into calculating the largest connected component in terms of number of included precincts. All precincts that do not belong to the largest connected component are added to the repair list. The effect of this procedure is to ensure that the disconnected precincts to be repaired and assigned to new districts are minimal in number, preserving the larger mass of connected precincts to be in the district.

**Repairing.**   The main repair procedure is performed for a single district by `repairOne-District.checkDistrictContiguity()` produces a minimal list of precincts disconnected from the main body of the district. For each of these precincts a random neighbor with a different district is selected and the precinct is assigned to that neighbor's district.

If, after repairing all of the districts (that need repairing), every district is within the population tolerance and is contiguous, then a feasible plan has been discovered and the algorithm cleans up, writes a file to record the plan, and exits. If any district is not within the population tolerance, then `shiftWhile` is invoked on the plan and another round of shifting, repairing, and checking for contiguity ensues. This continues until a feasible plan is found.

**SFSR pseudocode.**   To summarize the overall SFSR procedure, Algorithm 1 provides a pseudocode formulation of the entire procedure.

**Algorithm 1**: Pseudocode of Seed-Fill-Shift-Repair Algorithm for Districting

```
Data: Geographical data: List of precincts with geometries
Data: Parameters: Population tolerance, Number of districts
(numDistricts)
Result: Districting Solution
begin
  Seed:;
  Randomly pick numDistricts and assign them to one (unique) district;
  Fill:;
  while unassigned precincts do
    randomly pick one district;
    add unassigned adjoining precinct to district;
  while not done do
    while not within population balance do
      Shift:;
      randomly pick precinct on border;
      pick a neighbor precinct in another district;
      if precinct population >ideal population and neighbor population
      <ideal population then
        switch precinct to neighbor district;
      else if precinct population <ideal population and neighbor
      population >ideal population then
        switch neighbor precinct to district;
      if solution is contiguous then
        return solution;
      else
        Repair:;
        for district in district list do
          repair district;
          check district contiguity;
      if solution is contiguous then
        return solution;
```

## Computational results

The main objectives of this paper, as indicated above, are (i) to present the SFSR districting algorithm in a broadly accessible form, including code that implements it, and (ii) to make the case for what SFSR essays to do, which is to approach redistricting from a constraint satisfaction perspective and to produce a plurality of feasible solutions that can or might then serve in subsequent deliberations. It is beyond the scope of the present paper to undertake extensive analytics on outputs from SFSR. It is also beyond the scope to run computational comparisons of SFSR with the algorithms discussed in "Related work," an aim which in any event is further impeded by the lack of publicly available code implementing many of these algorithms, even

for results submitted as part of redistricting litigation. This state of affairs is commented on by [82], and we join them in calling for "open and reproducible development of tools for redistricting."

Illustrative analytics, however, are in order and that is the subject of the present section. After providing additional implementation and code details, we present two types of analysis: First, we applied SFSR to five different U.S. states and calculated 1000 districting solutions each. We report several descriptive statistics for these states as well as run times of the algorithm. Second, to focus on a particular state, we provide a deeper analysis of a much larger number of districting solutions for Pennsylvania and their visualizations.

## Implementation

SFSR is implemented in Python and uses a variety of standard packages such as pandas and Numpy. In addition, it relies on two packages specializing in geo-spatial analysis: PySAL (http://pysal.org), and GeoPandas (https://geopandas.org). These libraries afford fast calculation of geo-spatial information, checks for contiguity, and repairs for non-contiguous districts. In addition, our implementation is able to calculate multiple solutions simultaneously by (optionally) leveraging parallelization. Besides implementing the SFSR algorithm, the code also provides a module that communicates with the Census API provided by the U.S. Census Bureau via the Cenpy (https://pypi.org/project/cenpy/) Python package. This module supports automatic download of shapefiles and demographic American Community Survey (ACS) data for a specified U.S. state. The shapefiles and demographic information can then be used to create districting solutions.

For the subsequent analyses, the solutions were created on a workstation using an Intel Xeon Processor with 12 cores and a total of 128 GB RAM, although the program itself uses less than 1 GB of memory for the problems considered here. While the run time results presented here represent runs on the workstation, we saw similar run times with the same code on standard laptop/desktop configurations (Mac, Windows, and Linux) and thus do not expect a substantial run time increase when using a different setup.

## Congressional districting for U.S. states

To show that SFSR is able to create a plurality of solutions in a short amount of time, we created 1000 districting solutions for five US States: Arizona, Michigan, North Carolina, Pennsylvania, and Wisconsin. Arizona is one state that uses an independent commission to draw congressional districts; the other states have had maps thrown out by the courts on partisan gerrymandering grounds. (See *League of Women Voters v. Benson* 373 F. Supp. 3d 867 (E.D. Mich. 2019) (2019), *Harper v. Lewis* NO. 5:19-CV-452-FL (E.D.N.C. Oct. 22, 2019), and *League of Women Voters v. Commonwealth of Pennsylvania* 178 A.3d 737 (Pa. 2018) for the cases that overturned the congressional districts in these states. In Pennsylvania, for example, the map drawn by the state supreme court has no partisan bias [95]. The congressional district map in Wisconsin was not challenged in court. The legislative map in Wisconsin was found to be a partisan gerrymander (See *Whitford v. Gill* 218 F. Supp. 3d 837 (W.D. Wis. 2016). Simulated maps also demonstrate the legislative-drawn map was an outlier compared to a corpus of 200 reference plans [96]. The decisions in Michigan and Wisconsin were rendered moot by the U.S. Supreme Court decision in *Rucho v. Common Cause* 588 US __ (2019) that partisan gerrymandering presents a political question that is not justiciable by federal courts.) The number of Congressional districts to create for this problem follow the current number of Congressional districts, which are 9, 14, 13, and 8 for AZ, MI, NC, and WI, respectively. (We solved for 17 districts in PA, which presently has 18, in anticipation of a loss of one district

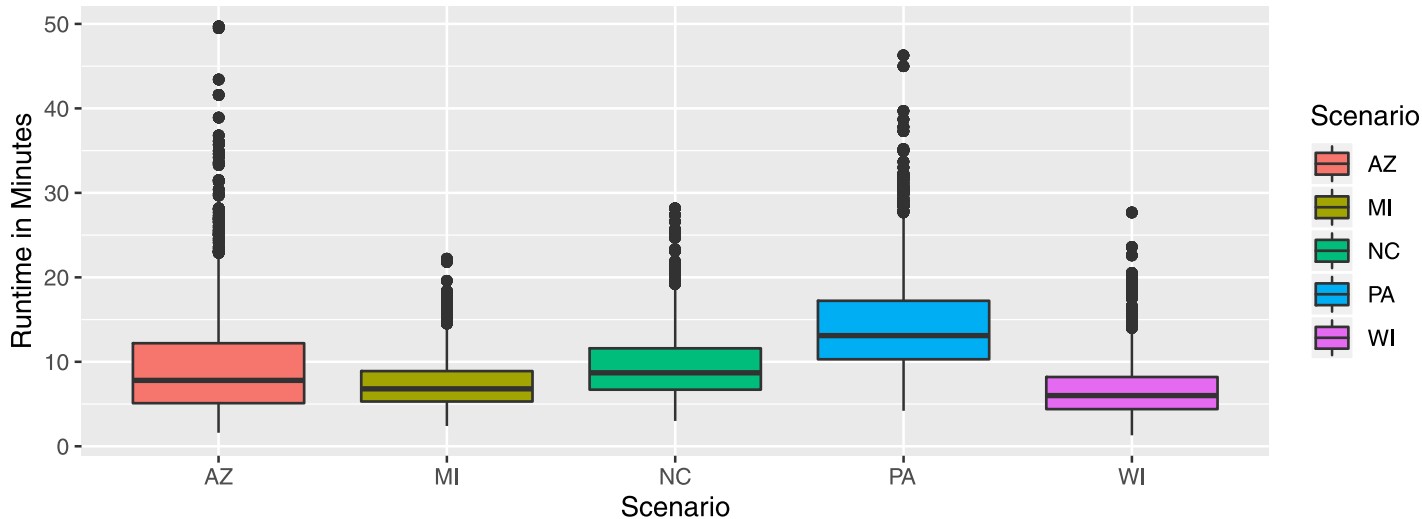

**Fig 1. Run time in minutes for districting plans based on the US State.**

after the 2020 census, see discussion below.) All solutions were created using a 1% population tolerance level, i.e., districts were acceptable if they are contiguous and within ±1% of the ideal population. For the underlying demographic population data we used the American Community Survey (ACS) 2018 data from the U.S. Census Bureau.

Fig 1 shows the run time in minutes to calculate one solution for each of the five considered U.S. states. The vast majority of solutions took less than 30 minutes to discover a plan. In fact, the average run times are 9.7, 7.4, 9.4, 14.4, and 6.7 minutes for AZ, MI, NC, PA, and WI solutions, respectively. All runs were completed within 50 minutes. Overall, the run time results are encouraging and show the feasibility of calculating a vast number of different solutions, in particular when parallelization is utilized. Considering the relative run time of the Seed-Fill-Shift-Repair procedures, on average the seed and fill part of the algorithm account for 1% of the run time and the shift and repair procedures account for almost 50% of the run time, respectively.

To analyze the created plans themselves, we look at three different measures that are often used in the analysis of districting plans. First, the number of split counties. Specifically, this metric investigates how many counties in the state have precincts in more than one Congressional district, i.e., counties that are split into more than one district. Second, the average deviation from the ideal population. While the tolerance of 1% guarantees that the smallest and largest districts are within ±1% of the ideal population, the average deviation from the ideal population considers the overall population balance of the plan. Third, the number of minority-majority districts. Specifically, for each plan we calculate the number of districts with more than either at least 50% or 75% white population as well as districts with more than 50% or 40% of Black population. This metric considers the fairness of the districting plans with respect to representation of minorities.

Considering the number of split counties and the average deviation from the ideal population, Fig 2 shows the absolute number of split counties, the relative number (percentage) of split counties, and the average deviation from the ideal population. With respect to the numbers of split counties, we see that there is considerable variety across the 1000 created plans. On average, the created plans resulted in 40%—65% of counties that are split into different districts. Increasing the number of created plans will make it more likely to find further solutions

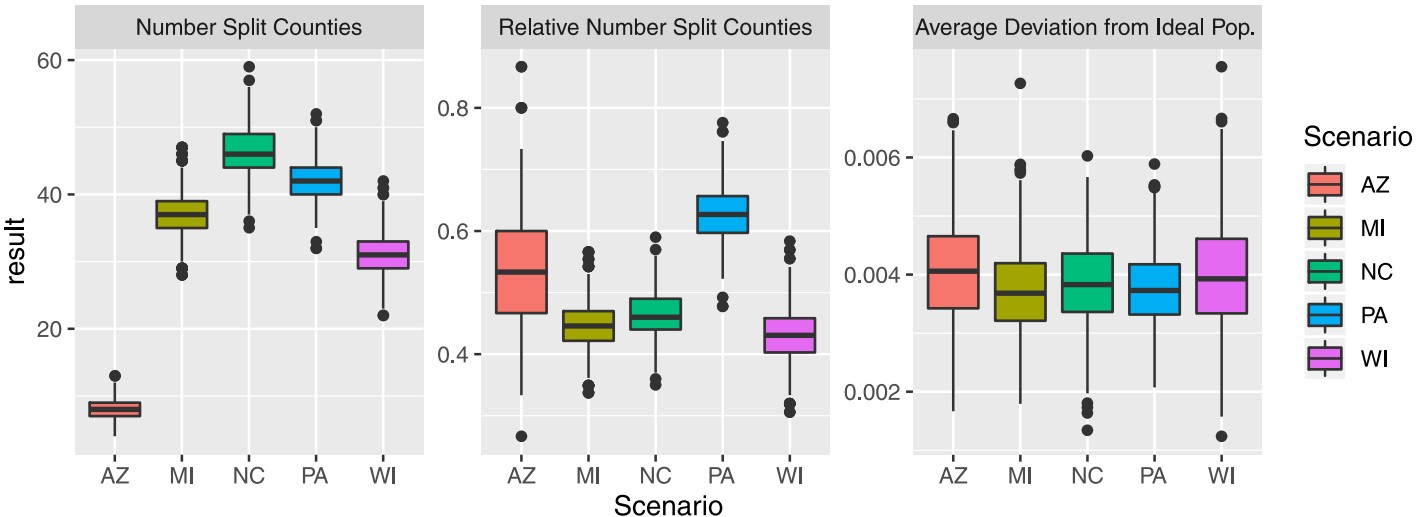

**Fig 2. Absolute and relative number (percentage) of counties split, and average deviation from ideal population.**

with a high or low number of split counties, the latter of which might be preferable for many voters.

Considering the average deviation from the ideal population, all plans yield a deviation of less than 0.8%, with the average plan having an average deviation around 0.4%. This means that on average, districts created by SFSR (with a 1% maximum tolerance) are within 0.4% of the ideal population. Districting plans are on average considerably closer to the ideal population than the allowed population tolerance.

Finally, to consider how different plans are representative of the demographic make-up of the voters, we consider the number of districts with a majority of white voters ("White Majority Districts" for districts with at least 50% whites, and "White Supermajority Districts" for districts with at least 75% whites). For Black voters we consider "Black Majority Districts" (districts with at least 50% Blacks), and "Black Opportunity Districts" (districts with at least 40% Blacks). For Latino voters we consider "Latino Majority Districts" and "Latino Opportunity Districts" for districts with at least 50% or 45% Latino voters respectively. Fig 3 shows the corresponding distributions for the plans created for the five states. We can observe several interesting results. First, similar to the range in the number of split counties, we see that the created solutions differ based on the number of minority majority districts. Second, in states with a higher percentage of minority voters (e.g., Black voters in MI, NC, PA and Latino voters in AZ) we find districting solutions where one or more districts are composed of more than 40% or 50% of voters from this minority group. For example, several districting solutions for MI and PA have 2 districts with more than 40% Black voters, and most solutions for AZ have between 1 and 4 districts (out of 9 total districts) with 40% or more Latino voters. This is an interesting result as the number of minority majority districts can be an important selection criterion in a subsequent decision process to select a final districting plan.

The information on minority and majority districts can also be used to analyze how representative a districting plan is with respect to the population demographics of a state. Table 5 provides general demographic information per state. For the earlier example of Arizona, a plan with 2 Latino Majority districts would correspond to 2 out of 9 districts having a Latino Majority, as compared to an actual percentage of 31.1% Latino population in AZ. Similarly, some districting plans for AZ have 8 districts with a White Majority (out of 9 total districts),

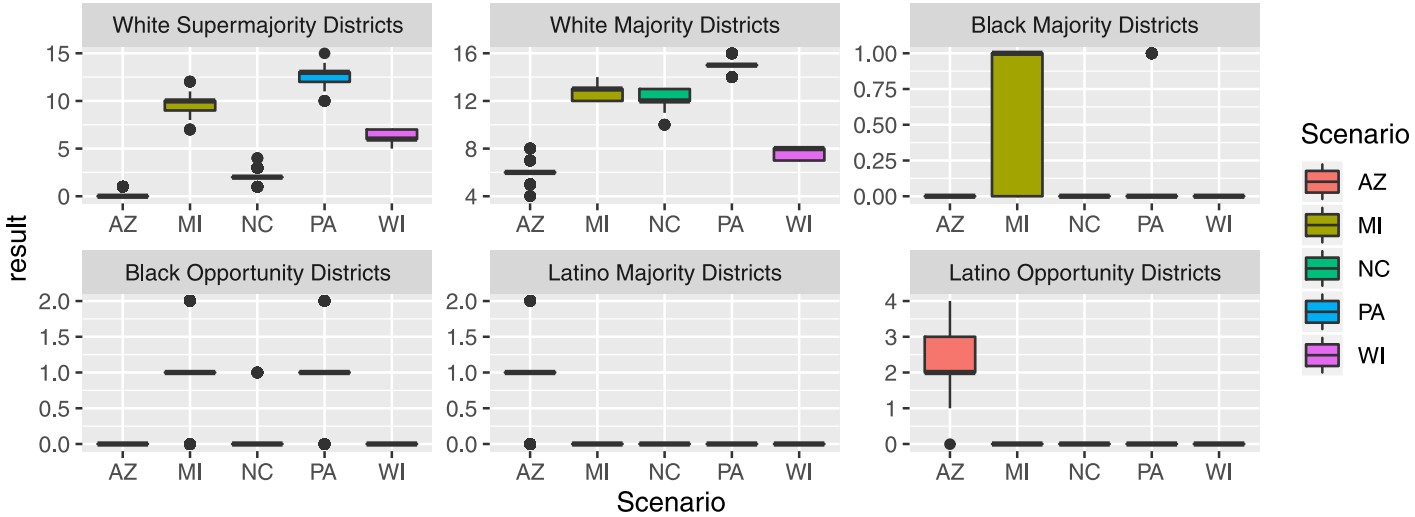

**Fig 3. Minority majority metrics for the created plans.**

which is over-proportionally more than the average 55.1% white population in AZ. A similar analysis can be run for other states. For example, for Michigan (MI) some plans have 1 Black Majority district (out of 14), compared to the 13.7% Black population in MI.

After this general analysis of the 1000 districting solutions for each state we will now discuss what we find in a corpus of 11,206 solutions created for Pennsylvania.

## Pennsylvania: Analysis of a larger corpus

We focus on the Commonwealth of Pennsylvania and use the American Community Survey (ACS) 2017 census tract data from the U.S. Census Bureau. (To be precise, we collected data for each census tract in Pennsylvania from Social Explorer, using the 5-year sample ending in 2017.) According to this source, the population of Pennsylvania is estimated to be 12,790,505. Anticipating that Pennsylvania will lose one Congressional district after the 2020 census (Apportionment estimates starting in 2014 suggested Pennsylvania is likely to lose a Congressional district, see the apportionment reports released by Election Data Services, Inc. See https://www.electiondataservices.com/wp-content/uploads/2014/10/NR_Appor14bwTablesMaps.pdf), we assume 17 districts in constructing our plans. The ideal population of a district is in consequence 752,383. We further assume a 1% population tolerance, so that a population feasible district has a population of 752,383 ± 7,524. Census Tract 9900, Erie County, Pennsylvania is the largest tract in terms its population, which is estimated to be 12,682. In fact, of the 3,218 census tracts in Pennsylvania, the ACS estimates that 126 have populations greater than the 1% tolerance of

**Table 5. Overview of congressional districts and white/ non-white population per considered state.**

|  | AZ | MI | NC | PA | WI |
|---|---|---|---|---|---|
| Number of Districts | 9 | 14 | 13 | 17 | 8 |
| Percentage White Population | 55.1 | 75.2 | 63.3 | 76.8 | 81.5 |
| Percentage Non-White Population | 44.9 | 24.8 | 36.7 | 23.2 | 18.5 |
| Percentage Black Population | 4.1 | 13.7 | 21.1 | 10.6 | 6.3 |
| Percentage Latino Population | 31.1 | 5 | 9.2 | 7.1 | 6.7 |

7,524, so that adding or removing any one of them from a population feasible district will make the district population infeasible.

The population slack condition is not satisfied for our data set, given a 1% tolerance. We nevertheless generated 11,206 plans without the algorithm ever failing to find a feasible solution within a small multiple (2 or 3) of the typical time for a run, which is about 10–12 minutes on a contemporary laptop computer. We used a variety of such computers to generate the corpus of 11,206 plans, both Windows machines and Macintosh machines, using the same code on each machine.

We turn now to evaluation of two important attributes of Congressional redistricting plans: county splits and minority-majority and minority-opportunity districts. Maintaining the integrity of places, particularly administrative divisions such as cities, school districts, and counties, is an often recognized desideratum for redistricting plans [44, 45]. Exploratory interviews and *amicus* court filings [97] suggest that for many people integrity of places ranks high in qualities they value in any redistricting plan. Small communities of interest often want to be concentrated in a small number of Congressional districts, the better to be able to influence their representatives. Parents often want their school districts to reside in single districts at the state level for the same reason [88, 98]. Respondents in exploratory interviews often express a desire to be represented together with the people they live and interact with most often, leading to a valorizing of compactness and integrity of places. The number of county splits in a Congressional districting plan is thus a measure of these values, with the smaller the number the better.

County splits can be counted in several different ways. We chose to report the number counties in a plan that reside in more than one Congressional district. As such, a county split into four Congressional districts adds only one to the county split total for the plan. There are 67 counties in Pennsylvania, capping the number of splits possible by our measure. The minimum number of splits is three because there are three counties whose populations exceed that of a feasible Congressional district including population slack: Philadelphia, Montgomery, and Allegheny. In the corpus of 11,206 plans found by SFSR, 450 plans have 30 or more unsplit counties, with a maximum unsplit count of 35 attained by 3 plans. See the "Supporting infomation" materials for a Jupyter Notebook for calculating county split information from the repository of plans.

Minority friendly districts, our second criterion, is important for legal and policy reasons under the Voting Rights Act and for moral reasons generally. Minority (non-white) citizens have historically often been the targets of discriminatory gerrymandering, aimed at reducing their electoral influence.

Table 6 shows the population minority distribution as estimated by the 2017 ACS. The non-white portion of the population is 22.7%, which would yield in expectation 3.87 of the 17 districts. The white portion alone is 77.2%, or 13.1 districts in expectation. The Black portion alone is 10.6%, or 1.8 districts in expectation.

Table 7 tells us much about the potential of the plans in the repository to address issues of racial balance. Blacks by their proportion of the population should have a preponderance of voters in 1.8 districts. In no plan is the number of Blacks more than 50% in more than 1 district, and more than 40% in more than 2 districts. The repository, however, contains 727 plans

**Table 6. Pennsylvania population categories from the ACS 2017 survey.**

| Total population | White | Black | Other | Latino |
|---|---|---|---|---|
| 12,790,505 | 9,881,135 | 1,358,263 | 676,274 | 874,833 |

**Table 7. Statistical summary of white and Black proportions in discovered districting plans.**

|  | W≥75 | W≥50 | B≥50 | B≥40 |
|---|---|---|---|---|
| mean | 12.681 | 15.1031 | 0.0649 | 1.070 |
| std | 0.765 | 0.397 | 0.246 | 0.657 |
| min | 9 | 14 | 0 | 0 |
| 25% | 12 | 15 | 0 | 1 |
| 50% | 13 | 15 | 0 | 1 |
| 75% | 13 | 15 | 0 | 2 |
| max | 15 | 17 | 1 | 2 |

Summary statistics comparing racial composition in the corpus of 11,206 plans. W≥75: across all 11,206 plans the mean number of districts per plan with at least 75% whites, the standard deviation of this number, the minimum, the maximum, and the quantile values. Explicitly, the average number of districts per plan that held at least 75% whites was 12.681, out of 17 districts per plan. In the 11,206 plans, the smallest number of districts with at least 75% whites in a plan was 9, the largest 15. The interpretation applies to the other columns as well. W≥50: across all plans, statistics on the number of districts with at least 50% whites. B≥50: across all plans, statistics on the number of districts with at least 50% Blacks. B≥40: across all plans, statistics on the number of districts with at least 40% Blacks.

in which Blacks have 1 district with 50% or more citizens and 2840 plans in which Blacks have either 1 or 2 districts with 40% or more citizens. Of the 11,206 plans, 9,153 have at least one district that is 40% Black or more.

Fig 4 is an example plan in which there is a single district that is Black majority and a second district with more that 40% Black residents.

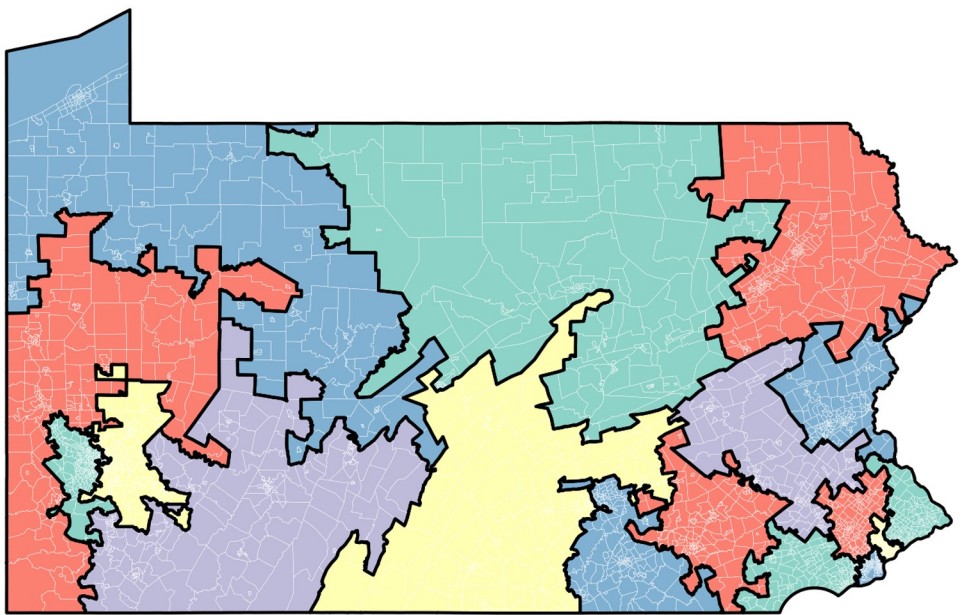

**Fig 4. PA plan with one Black majority district and a second district with more than 40% Black residents.** The small blue-gray district in the southeast of Pennsylvania, on the border, with a yellow district to its northeast, is 50.4% Black. It is split between Philadelphia County and Delaware County. The yellow district adjacent is 40.5% Black. It is split between Philadelphia County and Montgomery County. The plan is contained in file df2019-10-25x21x16x16x753018pc0.01x615.csv, which is part of the repository of 11,206 plans found.

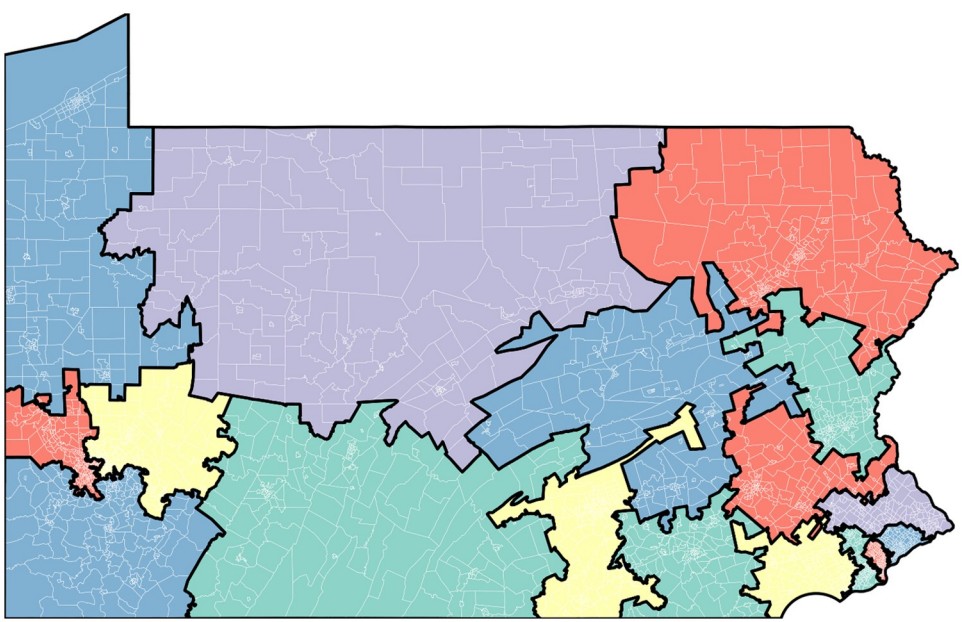

**Fig 5. PA plan with two districts with more than 40% Black residents, a third district with 53.1% non-whites, and 35 county splits.** The red district in the southeast on the southern border has a Black population that is 41.2% of the district's total. The green district to the west and on the southeast border is 40.8% Black. In addition, the blue district to the east of the red district is 53.1% non-white. All other districts have a white majority of at least 60%. Philadelphia County is split into these three districts. The plan is contained in the file df2020-01-31x14x56x15x610108pc0.01x1618. csv, which is part of the repository of 11,206 plans found.

Finally in our illustrative discussion of plan analytics with SFSR, we can examine the trade-offs available between county splits and minority friendly districts. By querying the plan repository we find that there are 115 plans in which the number of county split is at least 30 and in which there are 2 districts with a Black population of at least 40% of the district. Fig 5 shows a plan with two districts with more than 40% Black residents, a third district with more than 50% non-whites and 35 county splits, representing a good compromise solution found by SFSR in the corpus of 11,206 plans.

## Effect of solution set size

The previous analysis shows that solutions with a variety of properties can be found by SFSR. To analyze how the number of solutions affects the distribution of solution properties, we analyzed the previously described metrics for a different number of calculated solutions using the 11,206 Pennsylvania solutions as baseline.

Fig 6 shows the distribution of absolute and relative number of split counties as well as deviation from ideal population for different number of solutions. In general, we can see that calculating more solutions increases the number of solutions with small or large numbers for the respective metric. For example, the minimum number of split counties when 1000 solutions were calculated was 37, whereas this minimum number reduced to 32 when 11,206 solutions were calculated. While the majority of solutions fall within a consistent range, the extreme solutions become more likely. (Based on the current congress districts drawn using Census 2010 data, PA is divided into 18 congressional districts. The number of split counties is 14, and the population deviation 0.02. However, as our maps use the latest available demographics and thus calculate 17 districts, these numbers are not directly comparable to our created solutions.)

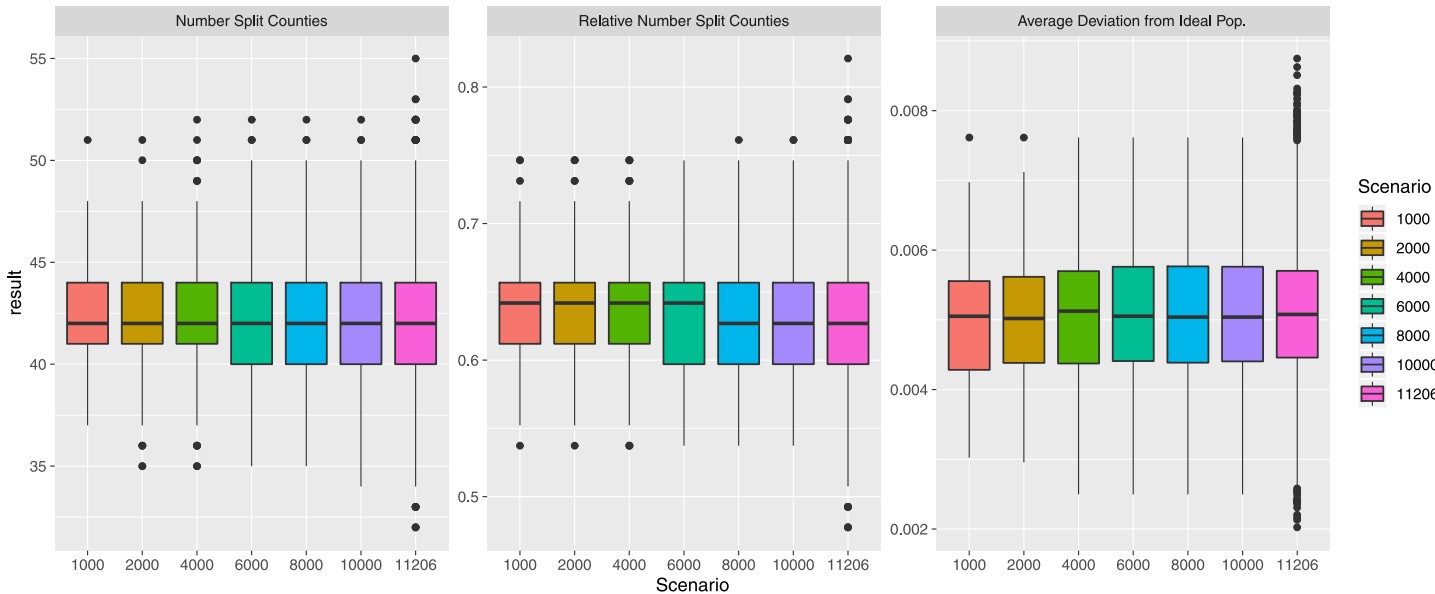

**Fig 6. Absolute and relative number of counties split and average deviation from ideal population for different number of solutions.**

Similarly, Fig 7 shows the distribution of minority-majority districts based on the solution set size. While the distributions here are more stable, we can see that increasing the number of calculated solution also leads to finding solutions with a high or low White Supermajority metric, for example.

SFSR has been demonstrated, in the examples discussed in this section, to be reliable in finding feasible redistricting plans. Each run of the algorithm (fixing the state and underlying data, here based on census tracts and Congressional districting) is an independent experiment that tests whether the algorithm will find a feasible plan within a multiple of about three times

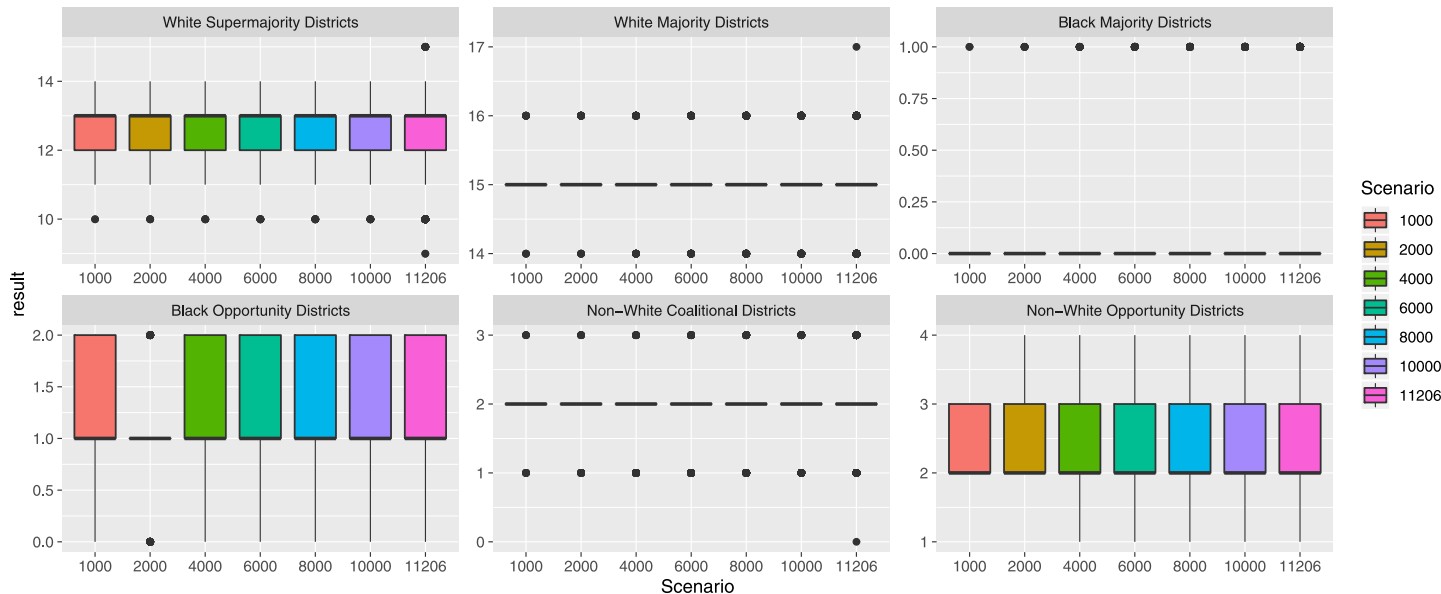

**Fig 7. Minority majority metrics for different numbers of calculated solutions.**

the average time to success. None of our experiments (1,000 runs for each of the five states and 11,206 runs for Pennsylvania) failed to find a feasible solution in a reasonable amount of time (as reported above). For a given state, if the probability of success of a run is as low as 0.99, the probability of undertaking 1,000 runs without failure is $0.99^{1000} \approx 4.3 \times 10^{-5}$. Similarly, the probability of undertaking 11,000 runs for Pennsylvania without failure if the probability of success is as low as 0.999 is $0.999^{11206} \approx 1.35 \times 10^{-5}$. The joint probability of no failure across all of these experiments is astronomically small. As noted above in our discussion of the population slack condition, SFSR may fail to find feasible solutions if the basic units, the precincts, are larger than the population slack value. Although some precincts do have population exceeding the population slack in our examples, SFSR nonetheless never failed to find a feasible solution. We can safely conclude that SFSR reliably finds feasible solutions, at least for the states tested and the conditions they were tested on. Moreover, we have identified an underlying cause of failure: too many precincts being larger than the population slack value, which we certainly observe if that value is set too low. The slack value of 1% of ideal population size, which we use in this paper, is the conventional value for redistricting studies. Given that Congressional districts are close in population across most states and given that census tracts have an average of about 4,000 people across all states, the results for the five states we report here are likely to obtain in many other states.

## Discussion

To summarize, the main purpose of this paper has been to present the SFSR (Seed-Fill-Shift-Repair) algorithm and to discuss why it should (as a constraint satisfaction algorithm), and how it can, be used in districting problems, in order to introduce it to communities interested in political redistricting. As part of this, we are making the code available for researchers so that more people can add to the body of research on redistricting. The "Methods" section describes the algorithm in detail and commented source code is available in the "Supporting information" S1 File. The "Computational results" section describes two example uses of SFSR. First, we use SFSR to obtain and analyze plans for five different U.S. States. Second, SFSR is used to create a corpus of 11,206 redistricting plans for Pennsylvania based on the 2017 American Community Survey data from the US Bureau of the Census. (These plans are available in a zipped folder of the repository of 11,206 plans, PA2017ACSPlans.zip, available from Zenodo: https://doi.org/10.5281/zenodo.3893078. In addition: Ancillary files for generating and processing PA districting plans with 2017 ACS data are available from Zenodo: https://doi.org/10.5281/zenodo.3893078; a Jupyter Notebook for calculating county split information, CountySplitsCalculator.ipynb, is available from Zenodo: https://doi.org/10.5281/zenodo.3911273; and a zipped folder of repositories of 1000 plans for each of 5 states, Five States 1000 AZ PA NC MI WI.zip, is available from Zenodo: https://doi.org/10.5281/zenodo.3931097.) The analysis is for purposes of illustration. We find a substantial diversity among the plans discovered by SFSR, as measured by two focal attributes of interest: county splits and minority friendly districts. The section serves to demonstrate how a large plurality of plans may be examined to find plans that score well on dimensions of import—county splits and minority friendly districts in our example—and to find plans that exhibit and accommodate values trade-offs among the dimensions of import.

A comprehensive analysis is beyond the scope of this paper, but the types of analysis displayed in the "Computational results" section are uniquely afforded by the presence of a large plurality of solutions, the corpus of 11,206 discovered plans in the present instance. Moreover, these analyses are essential components of any comprehensive examination of redistricting

alternatives, which is in the end about choosing in the presence of trade-offs among competing values attached to plans.

We now address several issues pertaining to SFSR, which arise in the larger context of redistricting. Districting problems, and political redistricting in particular, may be conceived abstractly as multi-objective constrained optimization problems. As we noted in the "Introduction" section, however, there is nothing close to general agreement regarding what the many objectives should be, what the most important ones are, or how competing objectives should be traded off. Criteria serving as objectives, moreover, are in many cases attended with uncertainty (e.g., in assessing partisan fairness based on past elections) and vagueness (e.g., in measuring compactness, or in trying to measure integrity of communities of interest, which measures should be used for these more general but vague concepts?). Finally, even if these obstacles were surmounted and a suitable multi-objective constrained optimization program were formulated, it is doubtful whether the ensuing computational complexity would permit much to be done by way of solving the model. Districting problems, recall, have long been known to be computationally complex.

The motivating idea of the SFSR algorithm is to circumvent these difficulties by eliminating the objective function(s), leaving a constraint satisfaction problem, which we have demonstrated to be tractable with SFSR in the cases of the five states discussed in the paper. We envision subsequent deliberation as a process along the following lines, which should be understood as a general template or pattern having indefinitely many variants.

1. Interested parties (redistricting commissions, legislative committees, civic groups, third sector activists, etc.) use SFSR (and/or alternative procedures) to generate a plan corpus, that is, a large number of redistricting plans whose districts are contiguous and in close population balance (presumably about 1%).

2. Interested parties apply any available methods to improve and otherwise modify the plans in the original corpus.

3. Interested parties identify criteria of import and assess each of a plurality of plans on the basis of these criteria.

4. Interested parties establish values and preferred trade-offs for the plans, then apply these value judgments to the plurality of assessed plans in order to identify, from the group's perspective, a number of most preferred plans.

5. All parties enter into a public discussion and negotiation in order to choose a single plan.

6. The chosen plan is modified at the margins to achieve a level of population balance acceptable to the courts and the plan is put into place.

Redistricting is inevitably, and properly, a values-laden endeavor. SFSR, by enabling the generation of large corpora of feasible plans, contributes to discovery of options and to transparency in the process of choice. We expect future research in this area will focus on generating preference orderings for the traditional redistricting criteria among target populations as a means to refine the algorithmic searching for plans that warrant consideration by public bodies.

## The question of random plans

We consider here a question touched upon briefly in the the "Related work" section: Can redistricting algorithms produce a random sample of the possible plans? The question arises in contested redistricting proceedings. A number of authors, e.g., [72, 77], have produced large

corpora of Congressional redistricting plans for a given state and suggested that these plans constitute a random sample from the space of possible plans. They then go on to argue that a given plan, produced by an apparently partisan legislative process, constitutes an illegal gerrymander because it departs strongly on an attribute of legal interest (minority friendly districts, partisan balance, etc.) from the distribution of the attribute in the corpus of generated plans. The argument is a statistical one: because a random draw from a random collection of the plans is highly unlikely to produce a plan as extreme on the attribute in question, it is plausible to assume that the plan was created intentionally to have such an extreme value on the attribute. It is not an innocent side-effect of an otherwise valid and legitimate process.

Our concern is not with the argument per se, although we find it problematic, but with the underlying assumption that the plan generation algorithms produce random samples that contain enough maps so that some maps capture desired features of the districts.

The two papers under discussion [72, 77] are focused on constructing probability distributions of plan characteristics. However, because their algorithms include repairs for population balance and either restrict adjustments for achieving population balance to maintain contiguity [72] or engage in a subsequent repair for contiguity [77] the resulting graphs do not constitute purely random samples.

SFSR has the greatest differences with the Chen and Rodden algorithm [72]. There, the first step is to build districts by first labeling every political unit as a district. Then they randomly select a district, list all adjacent districts, select the adjacent district with its centroid closest to the centroid of the originally selected district and combine this adjacent district with the originally selected district, repeating this process until the number of districts is reduced to the legally specified number of districts. This process is analogous to the formation of planets in the solar system, with dust particles combining with other dust particles to make rocks; rocks and dust combining to make planetoids; and dust, rocks and planetoids combining to make planets, leading to the size differences of Mercury and Jupiter. They then achieve population balance by moving adjacent political units from one district to the other as follows. They find the two adjacent districts with the biggest population difference, find the units on the adjacent borders from the larger district that can be moved while retaining contiguity, select from this set the unit with the centroid closest to the centroid of the smaller district and move that unit from the larger district to the smaller district.

Liu, Tam Cho, and Wang [77] have two methods for producing the initial districts prior to satisfying population constraints. Their first is close to SFSR. They begin by randomly selecting units as districts until they have the required number of districts. They then randomly select from all units adjacent to a district and add that unit to the district until all units are assigned to a district. In their second method they randomly select units as districts from units on the border and then proceed to add units as before. Starting from units on the border guarantees that no precinct in the interior of the district is surrounded by just one other district (a hole). They then have a more elaborate algorithm for shifting units to achieve population balance which involves shifting whole sequences of units rather than one at at time. They produce two hundred maps this way and then breed new solutions. The breeding process consists of taking two maps and constructing a map with every district an intersection of one district from one map and one district from the other map. This results in too many districts. Two adjacent ones are then randomly selected and combined into one district until the number of districts matches the required total. The population repair is then done.

SFSR resembles the first method of Liu, Tam Cho, and Wang for forming districts before balancing the population. We do single-unit swaps like Chen and Rodden. However, unlike Chen and Rodden, we select two adjacent districts with one being above the desired population and the other being below, then randomly selecting a unit on the joint boundary from the

larger district and shifting it to the smaller district We repeat this process until population balance is achieved. Contiguity is then checked and we apply a contiguity repair process. Chen and Rodden do a contiguity check before moving.

The fundamental point, in our view, is that *none* of these algorithms can produce a random sample from the space of redistricting plans that is sufficiently large to contain desirable maps to be used in redistricting debates.

To illustrate this, assume a square state that is divided into 16 square districts with equal population and no split districts. Also assume each state has 8 precincts on every border with every other state. Note there are 24 borders between the districts. A new map is created by swapping the same number of precincts on each border, randomly swapping 0 to 8 precincts per border. The resulting number of maps generated is 4.26467 E+98. If a billion maps are generated from this set of maps, the probability of generating the map without splits is 1/4.26467 E+86, a minuscule number. Yet, a human would find the map with no splits almost immediately. Consequently, the maps have to be treated as raw material for finding useful maps, not a definitive description of possible maps.

We propose a different goal. We want to generate a collection of maps that are usable either as starting points for building good maps for a political entity, e.g., a state or county, or be a pool of maps from which interesting ones can be selected. Our algorithm addresses the the problem of meeting the requirements for population balance and contiguity and it does not generate a purely random sample of maps, for the portion of the algorithm that achieves population balance would be inappropriate for random generation. This approach, of producing a large plurality of valid redistricting plans, facilitates adding other algorithms for modifying plans and generating maps that are good in additional criteria, such as compactness and competitiveness.

Each and all of these algorithms can be used to generate into a corpus of redistricting plans (for a given polity). They should be used for this purpose precisely because they do not produce random samples but each in its own way explores part of the space. The question is one of finding good plans. To that end it will be imperative, and there is no alternative but, to explore the productiveness of different biased approaches, both individually and in combination.

We are publishing our code to advance the goal of creating freely available algorithms that a community of researchers can use for creating corpora of plans that can be explored for finding attractive concrete possibilities and for adding their own algorithms for generating maps that improve upon the plans at hand. To illustrate, because SFSR does not address compactness, the maps it generates are typically ragged around the edges. An algorithm to smooth the edges can be added to post-process the plans in a corpus. This work is underway but is beyond the scope of the present paper.

## Conclusions

We conclude with two comments about how we envision improving and deploying this algorithm in future redistricting cycles.

If we think of this algorithm as a study in operations research, our future work can proceed by expanding the set of criteria and by exploring heuristics that start with feasible plans and improve them with regard to stated criteria. We can incorporate indicators for additional redistricting criteria (e.g., measures of compactness, partisan advantage, communities of interest) in order to move toward conceptualizing redistricting as a multi-attribute or multi-objective optimization model. Under this framework, our goal is not to maximize any given criterion but rather to develop alternative constraint satisfaction algorithms and other heuristics for

generating large pluralities of plans that satisfy a minimum threshold of all of these criteria and are able to find plans that score well on criteria of interest.

Under the logic of a multi-attribute utility framework, the relative importance of each criterion in the model can vary. For example, we might envision assigning population balance an importance greater than the importance assigned to compactness. The specifications of these weights lead us to our second concluding point that this algorithm can change the character of the redistricting process. The redistricting process is well-known to be dominated by insiders. This algorithm, in virtue of producing a plurality of solutions, would allow private citizens and interest groups to propose consideration sets of district maps to relevant redistricting authorities as a means to articulate a claim that links geographic units to political representation. In so doing, we can introduce an element of democratic participation into a structure of republican government.

## Supporting information

**S1 File. A zipped folder of a complete setup (files and folders) for running SFSR on the Pennsylvania data described in the paper, including instructions.** This file is also available via Zenodo: https://doi.org/10.5281/zenodo.3965006 (DOI 10.5281/zenodo.3965006).
(ZIP)

## Acknowledgments

All interpretations and conclusions are the responsibility of the authors and do not reflect the institutional positions of, among others, the Brennan Center for Justice at New York University School of Law. Preliminary discussion of this paper was presented by Steven Kimbrough and Peter Miller at the Joint EURO/ALIO International Conference on Applied Combinatorial Optimization, Bologna, Italy, in 2018. Kimbrough and Miller presented a related exploratory study at the annual meeting of the American Political Science Association, San Francisco, CA in 2015.

## Author Contributions

**Conceptualization:** Christian Haas, Lee Hachadoorian, Steven O. Kimbrough, Peter Miller, Frederic Murphy.

**Data curation:** Christian Haas, Lee Hachadoorian, Steven O. Kimbrough.

**Formal analysis:** Christian Haas, Steven O. Kimbrough.

**Investigation:** Steven O. Kimbrough, Peter Miller.

**Methodology:** Christian Haas, Lee Hachadoorian, Steven O. Kimbrough, Frederic Murphy.

**Software:** Christian Haas, Steven O. Kimbrough.

**Validation:** Christian Haas, Steven O. Kimbrough, Peter Miller.

**Visualization:** Lee Hachadoorian.

**Writing – original draft:** Lee Hachadoorian, Steven O. Kimbrough, Peter Miller, Frederic Murphy.

**Writing – review & editing:** Christian Haas, Lee Hachadoorian, Steven O. Kimbrough, Peter Miller, Frederic Murphy.

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
