## [Decision Letter · Decision Letter 0]

12 May 2020

PONE-D-20-09788

Seed-Fill-Shift-Repair: A  redistricting heuristic for civic deliberation

PLOS ONE

Dear Professor Kimbrough,

Thank you for submitting your manuscript to PLOS ONE. After careful consideration, we feel that it has merit but does not fully meet PLOS ONE’s publication criteria as it currently stands. Therefore, we invite you to submit a revised version of the manuscript that addresses the points raised during the review process.

We recommend that it should be revised taking into account the changes requested by the reviewers. Since the requested changes includes Major Revision, the revised manuscript will undergo the next round of review by the same reviewers.

We would appreciate receiving your revised manuscript by Jun 26 2020 11:59PM. To enhance the reproducibility of your results, we recommend that if applicable you deposit your laboratory protocols in protocols.io, where a protocol can be assigned its own identifier (DOI) such that it can be cited independently in the future. For instructions see: http://journals.plos.org/plosone/s/submission-guidelines#loc-laboratory-protocols

We look forward to receiving your revised manuscript.

Kind regards,

Baogui Xin, Ph.D.

Academic Editor

PLOS ONE

Journal Requirements:

3. Please upload a copy of Supporting Information Files which you refer to in your text on page 19.

Reviewers' comments:

Reviewer's Responses to Questions

**Comments to the Author**

1. Is the manuscript technically sound, and do the data support the conclusions?

Reviewer #1: Partly

Reviewer #2: No

2. Has the statistical analysis been performed appropriately and rigorously? 

Reviewer #1: No

Reviewer #2: No

3. Have the authors made all data underlying the findings in their manuscript fully available?

Reviewer #1: No

Reviewer #2: Yes

4. Is the manuscript presented in an intelligible fashion and written in standard English?

Reviewer #1: Yes

Reviewer #2: Yes

5. Review Comments to the Author

Reviewer #1: The paper suggests a heuristic called SFSR based on constraint satisfaction for creating electoral redistricting plans for the state of Pennsylvania. The problem addressed is relevant and the suggested method adopts an interesting strategy for its solution. I have just a few items I would like to address before I am fully convinced that this article has been published on PLOS-ONE.

1. INTRODUCTION

For motivation, it is important to give a more detailed context of the problem. To this, it would be useful to include a paragraph that briefly explains the functioning of the American electoral system, how the seats are divided in the representatives' house, as well the presentation of concepts associated with the theme (e.g. electoral precinct, gerrymandering ). As an example, the term gerrymandering (line 142) is used without its concept and implications being previously described. For a better understanding, a figure could illustrate examples of favoring situations caused by gerrymandering.

3. METHODS

As informed by the authors, the source code and results were not made available due to the size of the files. As a suggestion, the "ResearchGate" portal allows the creation of DOI and storage of files up to 512 MB.

The method was presented from the code perspective but as it is not available, its understanding was more laborious. For reproduction purposes, I think the most appropriate format would be the description of the algorithm in pseudocode, which contains the actions of the method in a generic way and independent of language or implementation strategy. Including a flowchart that represents the steps of the method would also assist in its comprehension. The details of the implementation are important and may be contained in a file that accompanies the code itself. It would be interesting if the files with input data (lines 429 to 432) were also made available.

For higher assertiveness, I see that it would be more appropriate for the detailing of the components described in the section "The SFSR algorithm in detail" to be done in the Methods section itself in the respective procedure. The operation of the "Contiguity Checking" procedure was not very clear and deserves further details, especially about the matrix reduction strategy described in lines 542 to 545.

COMPUTATIONAL RESULTS / DISCUSSION

I miss details about the development and experiment environment, with information about versions of the adopted language, libraries, operating system, and hardware specifications (basically processor and memory).

Although the authors gave an idea of the time consumed by the method (line 587), I think that this measure could be better explored. Considering that the method adopts repeated solution adjustment strategies, it would be interesting to describe information about the time spent in each step of the method, to have an idea of the implications related to the convergence time to the viable solution.

In the figures presented, it was not clear certain notes described in the text. For example, in Figure 1 (lines 613-614) it is not evident where splits occurred in the counties; as well as in Figure 2 (lines 632 and 633) the districts with the highest incidence of minorities. Highlighting the changes in the figure can make the benefits of the proposed method clearer.

Table 2 needs to be described in more detail, it was not clear to me what the percentages represent 25%, 50% and 75% (below min). As an example, in my understanding, 25% of the plans have 12 districts with the number of whites greater than 75% (W> = 75). Would it be this? I imagine that a scatter plot can give an overview of the characteristics of the plans obtained (the choice of metrics is at the preference of the authors).

The experiments were based on a single dataset. I think it is important to consider other datasets for greater variability of scenarios, for example, different states or years. An interesting analysis would be to compare a redistricting obtained by the SFSR with another that has adopted in a real circumstance.

At various points in the text, the authors make it clear that a more extensive analysis of the method is outside the scope of the work. However, I see that this item is essential to meet the scientific rigor required by PLOS-ONE. In the Discussion section, the authors raise an interesting question about random plans and the advantages of SFSR over other methods. As a way of validating what was discussed, I think it is important that the methods in question, or at least one of them, are compared with SFSR concerning the execution time and quality of the generated plan. To compare the quality of the plans, the authors could adopt one or more external measures of your choice, for example, the number of counties splits, compactness, and so on.

I think that is valid a discussion about the flexibility (or not) of the SFSR to include other types of requirements.

Reviewer #2: The present manuscript discussed on the used of Seed-Fill-Shift-Repair (SFSR) districting algorithm. However, there are few points that the authors may consider in order to improve this manuscript:

1. The abstract is too technical and not adhere overview of the problem. The authors had straightforwardly describe the method, which at some point could be good in some readers, but it would be better if the authors can explain the general problem that they need to solve;

2. The explanation about the method is very intuitive, which is very good for those who are familiar in this field like myself, but it would be better if the authors can present some other ways to present the proposed method (i.e. pseudocode, flowchart). This is very helpful especially when explaining the experimental setup

3. The results are very limited and need more proof i.e. using statistical analysis

6. PLOS authors have the option to publish the peer review history of their article (what does this mean?). If published, this will include your full peer review and any attached files.

Reviewer #1: No

Reviewer #2: No

---

## [Author Response · Author response to Decision Letter 0]

18 Jul 2020

We have posted an extensive "Response to Reviewers" which we think answers to every point raised. Also, we submitted a cover letter describing this.

---

## [Decision Letter · Decision Letter 1]

6 Aug 2020

Seed-Fill-Shift-Repair: A  redistricting heuristic for civic deliberation

PONE-D-20-09788R1

Dear Dr. Kimbrough,

We’re pleased to inform you that your manuscript has been judged scientifically suitable for publication and will be formally accepted for publication once it meets all outstanding technical requirements.

Kind regards,

Baogui Xin, Ph.D.

Academic Editor

PLOS ONE

Additional Editor Comments (optional):

Reviewers' comments:

Reviewer's Responses to Questions

**Comments to the Author**

1. If the authors have adequately addressed your comments raised in a previous round of review and you feel that this manuscript is now acceptable for publication, you may indicate that here to bypass the “Comments to the Author” section, enter your conflict of interest statement in the “Confidential to Editor” section, and submit your "Accept" recommendation.

Reviewer #1: All comments have been addressed

Reviewer #2: All comments have been addressed

2. Is the manuscript technically sound, and do the data support the conclusions?

Reviewer #1: Yes

Reviewer #2: Yes

3. Has the statistical analysis been performed appropriately and rigorously? 

Reviewer #1: Yes

Reviewer #2: Yes

4. Have the authors made all data underlying the findings in their manuscript fully available?

Reviewer #1: Yes

Reviewer #2: Yes

5. Is the manuscript presented in an intelligible fashion and written in standard English?

Reviewer #1: Yes

Reviewer #2: Yes

6. Review Comments to the Author

Reviewer #1: (No Response)

Reviewer #2: The present manuscript describes an interesting application of computational science for political scenario. It would be better if the authors could emphasize a bit more on the fundamentals of the algorithmic point-of-view, i.e comparative analysis of different other algorithms so that the readers able to see how the proposed algorithm performs over other algorithms.

7. PLOS authors have the option to publish the peer review history of their article (what does this mean?). If published, this will include your full peer review and any attached files.

Reviewer #1: **Yes: **Rafael de Magalhães Dias Frinhani

Reviewer #2: No

---

## [Editor Report · Acceptance letter]

12 Aug 2020

PONE-D-20-09788R1 

Seed-Fill-Shift-Repair: A redistricting heuristic for civic deliberation 

Dear Dr. Kimbrough:

I'm pleased to inform you that your manuscript has been deemed suitable for publication in PLOS ONE. Congratulations! Your manuscript is now with our production department. 

Kind regards, 

on behalf of

Professor Baogui Xin 

Academic Editor

PLOS ONE